# DNA binding drives the association of BRG1/hBRM bromodomains with nucleosomes

Emma A. Morrison[1], Julio C. Sanchez[1,*], Jehnna L. Ronan[2,*], Daniel P. Farrell[1,†], Katayoun Varzavand[1], Jenna K. Johnson[1,†], Brian X. Gu[1], Gerald R. Crabtree[2,3] & Catherine A. Musselman[1]

BRG1 and BRM, central components of the BAF (mSWI/SNF) chromatin remodelling complex, are critical in chromatin structure regulation. Here, we show that the human BRM (hBRM) bromodomain (BRD) has moderate specificity for H3K14ac. Surprisingly, we also find that both BRG1 and hBRM BRDs have DNA-binding activity. We demonstrate that the BRDs associate with DNA through a surface basic patch and that the BRD and an adjacent AT-hook make multivalent contacts with DNA, leading to robust affinity and moderate specificity for AT-rich elements. Although we show that the BRDs can bind to both DNA and H3K14ac simultaneously, the histone-binding activity does not contribute substantially to nucleosome targeting in vitro. In addition, we find that neither BRD histone nor DNA binding contribute to the global chromatin affinity of BRG1 in mouse embryonic stem cells. Together, our results suggest that association of the BRG1/hBRM BRD with nucleosomes plays a regulatory rather than targeting role in BAF activity.

[1] Department of Biochemistry, Carver College of Medicine, University of Iowa, Iowa City, Iowa 52242, USA. [2] Program in Cancer Biology, and Departments of Pathology and Developmental Biology, Stanford University School of Medicine, Stanford, California 94305, USA. [3] Howard Hughes Medical Institute, Stanford University School of Medicine, Stanford, California 94305, USA. * These authors contributed equally to this work. † Present addresses: Department of Biochemistry, University of Washington, Seattle, Washington 98195, USA (D.P.F.); Center for Cancer Research, National Cancer Institute, Frederick, Maryland 21702, USA (J.K.J.). Correspondence and requests for materials should be addressed to C.A.M. (email: catherine-musselman@uiowa.edu).

Gene expression profiles are extensively modulated throughout development and in the maintenance of somatic cells, and proper regulation of gene expression requires orchestrated changes in the physical state of chromatin. Two of the main mechanisms by which chromatin is regulated are covalent modification of histone proteins and DNA (chromatin modification) and ATP-dependent remodelling of nucleosomes (chromatin remodelling). Histone and DNA modifications can have a direct impact on chromatin structure or can act to recruit or regulate the activity of chromatin acting factors[1,2]. ATP-dependent chromatin remodellers canonically function by disrupting the contacts between histones and DNA, leading to sliding or conformational change of the nucleosome or eviction of histones. Chromatin modification and remodelling mechanisms are largely cooperative[3], leading to a complex regulatory network that, together with the action of activators and repressors, ultimately determines the levels of gene expression and cell identity.

The mammalian SWI/SNF (mSWI/SNF) or BAF (BRG1- and BRM-associated factor) chromatin remodelling complexes are large multi-subunit assemblies containing 13–15 subunits. The exact composition of these complexes is heterogeneous across the cell type, and this combinatorial assembly is thought to afford functional specificity[4]. BAF complexes have both gene activating and repressing activity through canonical remodelling mechanisms and are also implicated in DNA damage response and DNA decatenation[5–7]. The central ATPase of the BAF complexes is alternatively either BRG1 (brahma-related gene 1) or BRM (brahma), encoded by the *SMARCA4* and *SMARCA2* genes, respectively (Fig. 1a). These ATPases are >70% identical and display similar *in vitro* activity[8]. It has been found that they largely cannot substitute for one another in a cellular context, which may be due to differential association with transcription factors[9]. Subunits of the BAF complex are mutated in 20% of all human tumors[10], thus an understanding of the function of this complex is critical for understanding disease aetiology and in the development of specific inhibitors.

Early genetic studies in the homologous yeast complex (ySWI/SNF) revealed genetic correlation with histone acetyltransferases[11], and studies *in vitro* demonstrated that chromatin template targeting and remodelling activity is augmented by histone hyper-acetylation[12]. A conserved bromodomain (BRD) in the ATPase subunit has been implicated in mediating the response to histone acetylation (Fig. 1a)[13–15]. BRDs form a class of well characterized effector domains that recognize acetylated lysines, and several are important in targeting their host proteins or complexes, such as BRD4 and TIP5, to acetylated chromatin. In fact, BRD inhibitors have shown immense promise in controlling gene expression in disease, particularly in oncology[16]. As such, there has been tremendous interest in developing inhibitors of BRG1/BRM BRD[17,18]. However, reports have been contradictory regarding the importance of the mammalian BRG1/BRM BRD in BAF function. *In vitro* assays have shown that BRG1 is preferentially targeted to or retained on hyper-acetylated chromatin templates[19]. Deletion of the BRD reduced BRG1 recruitment of the REST repressor to target promoters in HEK293 cells treated with the TSA histone deacetylase (HDAC) inhibitor[20]. BRG1 association with acetylated nucleosomes is critical in DNA damage response, dependent on the BRD[21,22]. In addition, deletion of the BRM BRD led to a modest effect on the ability of BRM to bind nucleosome templates *in vitro* and to reverse the transformed phenotype in Ras-transformed fibroblasts. However, BRG1 BRD has only millimolar affinity for acetylated histone tails[23,24], and small molecule inhibition of BRG1/BRM BRD does not decrease BRG1/BRM chromatin association in cells unless pre-treated with HDAC inhibitors[17,18,25]. Therefore, the potential role of the BRD in targeting the BAF complex to acetylated chromatin and its therapeutic potential has not yet been resolved.

Here, we demonstrate that human BRM (hBRM) BRD binds all acetylated histone tails weakly but with moderate specificity for H3K14ac, identical to what has been observed for BRG1 BRD[23,24]. We further demonstrate a previously unrecognized interaction of these BRDs with DNA, which is further enhanced by the adjacent AT-hook motif through multivalent contacts. Finally, though we find that the BRDs can bind both DNA and H3K14ac tails simultaneously, histone-binding activity does not significantly contribute to BRG1 or hBRM BRD association with nucleosomes *in vitro*. In addition, we find that neither the histone nor DNA binding activity of the BRD contribute to the global chromatin affinity of BRG1 in mouse embryonic stem cells (ESCs). Our results support a model in which the BRG1/BRM BRD plays a regulatory rather than targeting role in BAF complex function.

## Results

**hBRM BRD has moderate specificity for H3K14ac.** Previous studies have shown that BRG1 BRD binds acetylated histone tails weakly, with moderate specificity for histone H3 acetylated at Lys14 (H3K14ac). Though peptide-based screens have suggested that hBRM BRD may have distinct specificity[26], its association with histone tails has not been thoroughly investigated. To assess the binding activity of hBRM BRD for histone tails, we utilized nuclear magnetic resonance (NMR) spectroscopy. $^1$H-$^{15}$N heteronuclear single quantum coherence (HSQC) spectra were recorded on $^{15}$N-labelled hBRM BRD apo and upon titration of singly acetylated histone peptides. Titration of all acetylated substrates led to chemical shift perturbations (CSPs) of BRD resonances, indicating binding (Fig. 1b). Plotting the CSPs as a function of BRD residue reveals that CSPs are centred around the ZA and BC loops, with the greatest CSPs observed for H3K14ac (Fig. 1b,c). Mapping the most significant CSPs onto a structure of hBRM BRD demonstrates that these residues form a well-defined binding pocket on one end of the alpha helical bundle, which is consistent with the canonical BRD acetyl-lysine-binding pocket. This is a largely hydrophobic pocket and includes the conserved F1427, L1430, F1481 and N1482 acetyl-lysine coordinating residues (Fig. 1d,e).

Dissociation constants ($K_d$) were calculated from normalized CSPs using a single-site binding model accounting for ligand depletion. $K_d$ values reveal tightest association with H3K14ac at $K_d = 0.90\,\text{mM} \pm 0.06\,\text{mM}$ (where the error is the standard deviation), which is dependent on the acetylation of Lys14. All other acetylated peptides bound over twofold weaker (Fig. 1f). However, titration with free Kac yields a $K_d > 30\,\text{mM}$, revealing that the histone context is important even in the weakest binding peptides. This data reveals that, identical to what has been observed previously for BRG1 BRD[23,24], hBRM BRD binds all acetylated histone peptides weakly with moderate specificity for H3K14ac. Though BRDs are usually observed to bind histone tails more weakly than other effector domains such as PHD fingers and Tudor domains[27,28], it is notable that BRG1/hBRM BRDs bind weakly even for BRDs, including those specific for H3K14ac, which have affinities in the range of $K_d \sim 16\text{-}500\,\mu\text{M}$ (refs 29–32). To determine whether the binding pocket is capable of accommodating multiple acetyl-lysines, an H3 peptide acetylated at K14,18,23,27 (H3Kac(quad)) was titrated into $^{15}$N-labelled hBRM BRD. The same trajectories and CSPs were seen for H3Kac(quad) as for H3K14ac (Fig. 1b,c). This indicates the BRD binds H3Kac(quad) with the same binding mode and likely does not accommodate multiple acetyl marks. However, the

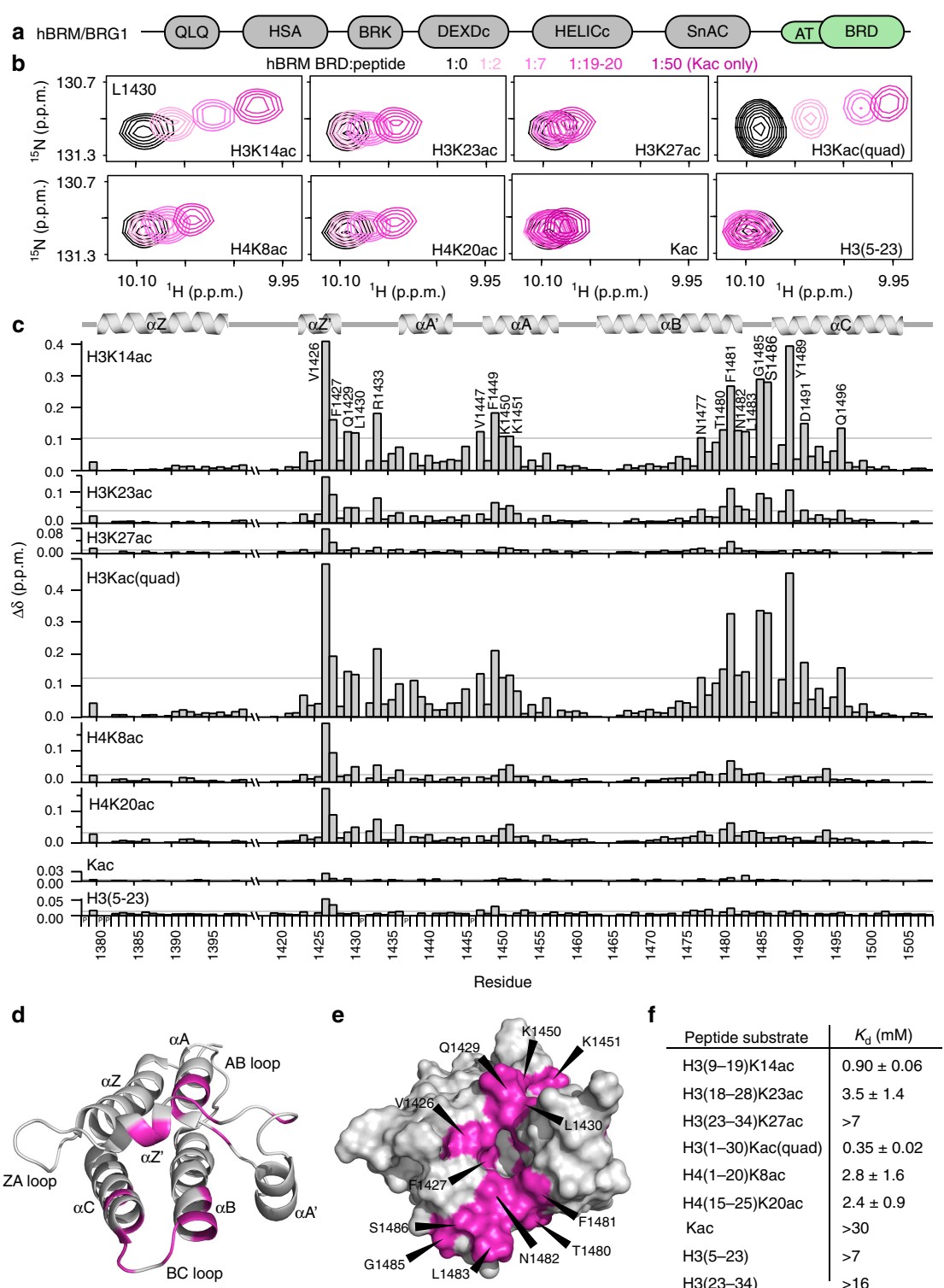

**Figure 1 | The hBRM BRD binds acetylated histone tails weakly with specificity for H3K14ac.** (**a**) The domain architecture of the hBRM and BRG1 ATPase subunits of the BAF complex. (**b**) Overlay of $^1$H-$^{15}$N HSQC spectra of $^{15}$N-hBRM BRD upon titration of histone peptide. The selected region displays the resonance for L1430, and spectra are colour coded according to protein:peptide molar ratio as shown in the legend. Peptide titrations were collected at protein:peptide molar ratios of 1:0, 1:0.5, 1:1, 1:2, 1:4, 1:7, 1:12 and 1:19-20. The free acetyl-lysine titration included the additional 1:50 point. For clarity, only 4-5 points are displayed. (**c**) Corresponding normalized CSPs plotted as a function of residue for each histone peptide ligand shown in **b** for the protein:peptide molar ratio of 1:19-20. The secondary structure of the BRD is denoted above the plots, and residues that were perturbed greater than the average plus two standard deviations after trimming off the top 10% of CSPs are labelled for binding to H3K14ac. A grey line marks this level of significance for each titration. (**d,e**) Structure of the hBRM BRD (PDB ID 2DAT) in cartoon representation with the secondary structure elements labelled (**d**) and surface representation (**e**). Residues that are significantly perturbed upon binding to H3K14ac are coloured in pink and labelled. (**f**) Dissociation constants ($K_d$) determined from NMR titrations with $^{15}$N-hBRM BRD. $K_d$ values are fit to a single-site binding model under ligand-depleted conditions using 8-9 titration points. Shown are the averages over individual residues with significant CSPs (12-19 in total) and associated standard deviation.

affinity increases nearly threefold, likely due to the increased concentration of acetyl marks (Fig. 1f).

A previous model of BRG1 BRD with the H3 tail suggested that specificity may arise from interaction with H3R17 and H3P16, which make a hydrogen bond with the backbone carbonyl of BRD T1538 and hydrophobic contacts with BRD L1494, respectively[24]. Both residues are conserved in hBRM (T1480 and L1436). Similar contacts were also observed to be important for interaction of Baz2B BRD with H3K14ac[31,32]. Two other peptides tested here, H3K23ac and H4K20ac, have a +3 Arg with hydrophobic residues at the +1 and +2 positions but bind more weakly than H3K14ac (Fig. 1f). As compared to the H3K14ac sequence, H4K20ac harbours two bulky residues at the +1 and +2 positions. Compared to the CSPs induced by H3K14ac, H4K20ac leads to significantly fewer CSPs around hBRM T1480 consistent with occlusion of the interaction with the +3 Arg (Supplementary Fig. 1). In contrast, H3K23ac, with Ala at the +1 and +2 positions, perturbs several residues around T1480, similar to H3K14ac (Fig. 1c and Supplementary Fig. 1) and suggesting an interaction between H3R26 and T1480. However, the weaker affinity for H3K23ac suggests a critical role of the +2 Pro of H3K14ac. The only other histone tail Lys with a +3 Arg is H4K16, which contains +1 Arg and +2 His, and consistent with this analysis, previous studies reveal minimal interaction with BRG1 BRD[23,24].

Together, our results reveal that hBRM BRD binds acetylated histone peptides with comparable affinity to those reported for BRG1 BRD[23,24], though these are weaker than that reported in ref. 27. Although overall the association of hBRM/BRG1 BRDs is significantly weaker than observed for other BRDs, both demonstrate specificity for H3K14ac.

**hBRM BRD isoforms have identical histone-binding activity.** hBRM is alternatively spliced, and two of these splice isoforms vary within the BRD[33]. These two isoforms differ by an 18 amino acid segment between the αZ and αZ′ helices (Supplementary Fig. 2a). The short form, sometimes referred to as hBRM-B BRD, is most highly homologous to BRG1 BRD (unless explicitly noted, any discussion of hBRM BRD refers to this short isoform). To determine the effect of the 18-aa insertion on the BRD fold and histone-binding activity, the longer BRD isoform (hBRM-A BRD) was examined. A comparison of $^1$H-$^{15}$N HSQC spectra for both isoforms reveals that the chemical shift values for residues in the additional 18-aa in hBRM-A fall between 8.0 and 8.6 p.p.m. in the $^1$H dimension, suggesting that the insert is unstructured (Fig. 2a). Chemical shift values for resonances corresponding to the remainder of the BRD overlay well between hBRM-A and hBRM-B, indicating the same overall fold. Compared to hBRM-B, CSPs and line broadening are observed in hBRM-A resonances corresponding to the very C-terminal portion of αZ and the first half of the ZA-loop (through αZ′) (Supplementary Fig. 2b), which flank the insertion. In addition, Phe1449 in αA is broadened and some residues in αC show small CSPs. The observed line broadening is suggestive of altered dynamics. Consistent with this, circular dichroism (CD) thermal melts reveal that the insert destabilizes the BRD, shifting the unfolding curve −20 °C (Fig. 2b). However, histone-binding activity is identical (Supplementary Table 1). Together, this demonstrates that the additional 18-aa in hBRM-A form a largely unstructured loop, which does not perturb the overall fold or histone-binding activity of the BRD.

**hBRM and BRG1 BRDs have DNA-binding activity.** Although it has been suggested that hBRM/BRG1 BRDs may be important for targeting acetylated chromatin, the weak affinity for acetylated

histone tail peptides brings this into question. One possibility is that the context of the nucleosome is necessary to recover the full binding activity. To test this, we ran electrophoretic mobility shift assays (EMSAs) of hBRM and BRG1 BRDs with the nucleosome core particle (NCP) reconstituted with Widom 601 DNA and devoid of any post-translational modifications. A strong shift in the position of NCP on the gel was observed upon increasing concentrations of either BRD, revealing a robust interaction even in the absence of histone acetylation (Fig. 3a, top). Surprisingly, EMSAs performed with DNA alone revealed a direct interaction with DNA (Fig. 3a, bottom). This is in contrast to a previous study of BRG1 BRD[34]. The reason underlying the different result is not clear, but is likely attributable to differences in sample preparation or experimental approach.

To further characterize this interaction, we utilized NMR spectroscopy. Two double-stranded (ds) DNA oligos representing fragments of the 601 sequence were tested (5′-CTCAATTGGT-3′ and 5′-CGTAGACAGCT-3′, referred to as DNAI and DNAII).[1] H-$^{15}$N HSQC spectra were recorded on $^{15}$N-labelled hBRM BRD upon titration of DNA. Addition of either DNA led to significant CSPs (Fig. 3b,d), demonstrating a robust association. Dissociation constants for were calculated from the normalized CSPs and yielded $K_d \sim 500\,\mu$M for both DNAI and DNAII (Fig. 3c, Supplementary Fig. 5 and Supplementary Table 2), suggesting a non-sequence-specific interaction. The residues exhibiting the greatest CSPs reside in αA, the ZA and AB loops, and the very N-terminal end of αZ. The binding pockets are nearly identical for both DNA substrates (Fig. 3d). Mapping these CSPs onto the structure of hBRM BRD reveals that the DNA-binding pocket coincides with a highly basic surface patch (Fig. 3e,f). To confirm the importance of these residues for binding, a triple mutant (K1450D,K1451N,K1460A) was generated. This mutant retained its fold and could still associate with H3K14ac peptide (Supplementary Fig. 8a), EMSAs revealed significant abrogation of binding to DNA and NCPs (Fig. 3a, right). The identity of residues in the DNA-binding pocket are identical in BRG1 excepting K1382, which corresponds to N1457 in BRG1. The hBRM-A BRD isoform was found to have modestly tighter DNA binding (Supplementary Fig. 4a and Supplementary Table 2), but resonances for residues in the 18-aa insertion are not perturbed upon addition of DNA (Supplementary Fig. 4c).

**The AT-hook and BRD bind DNA multivalently.** An AT-hook is found just a few residues N-terminal to the BRDs in both hBRM and BRG1 (Supplementary Fig. 3a). AT-hooks are small DNA-binding motifs found in a number of chromatin co-factors[35] that contain a core GRP sequence flanked by R/K-rich regions. They preferentially bind in the narrow minor groove formed by AT-rich DNA elements. hBRM and BRG1 AT-hooks have been shown to bind both linear and cruciform DNA, preferring linear[34,36], and hBRM AT-hook is critical for retention of hBRM at chromatin and its function in fibroblasts[36].

To investigate the effect of the AT-hook on the BRD interaction with DNA, we performed EMSAs using constructs containing both the AT-hook and BRD (AT-BRD) with 601 DNA, which notably contains three AT-rich regions (Fig. 4a, bottom and Supplementary Fig. 4b, top). Addition of hBRM or BRG1 AT-BRD led to a shift in the DNA, which starts at lower molar ratios than for BRD alone and results in a distinctive laddering pattern at low protein:DNA ratios, which was not seen with the BRD alone. EMSAs with the AT-BRD and NCP (Fig. 4a, top and Supplementary Fig. 4b, middle) are not as significantly different as compared to the BRD alone (Fig. 3a), although a more distinctive laddering pattern is also evident. The difference

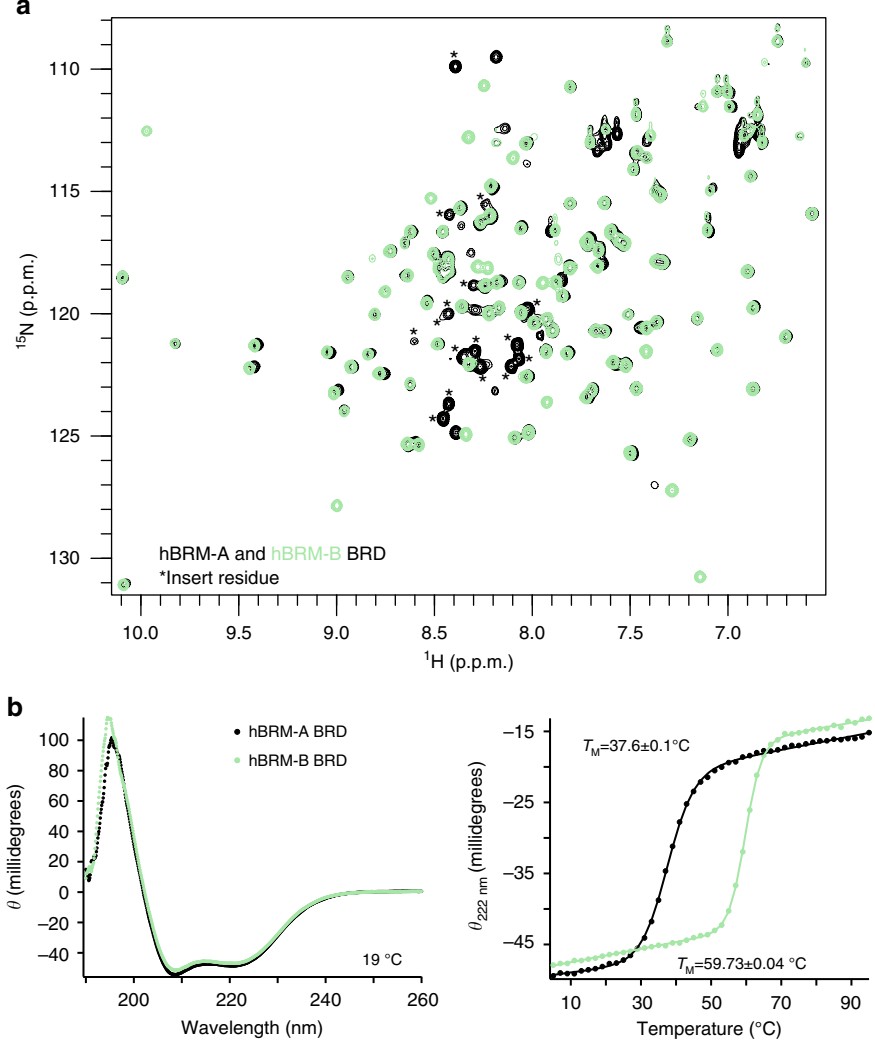

**Figure 2 | The hBRM BRD splice variants have the same overall fold but different thermal stability.** (**a**) Overlay of $^1$H-$^{15}$N HSQC spectra of $^{15}$N-hBRM-A (black) and $^{15}$N-hBRM-B (green) BRDs. Resonances corresponding to the extra 18-aa in hBRM-A are labelled (\*). (**b**) CD data collected on the two splice variants of the hBRM BRD. CD spectra collected at 19 °C on the hBRM-A (black) and hBRM-B (green) BRDs confirm an α-helical structure for both splice variants (left). Thermal denaturation curves are plotted as the raw ellipticity measured at 222 nm as a function of temperature and show that the hBRM-A BRD curve (black) is shifted by ~20 °C from the hBRM-B BRD curve (green) (right). The thermal denaturation curves were fit to the Gibbs–Helmholtz equation for unfolding as a function of temperature (solid lines), and the fit values along with the error in the fit for the melting temperatures ($T_M$) are shown.

between free DNA and nucleosome may be due to changes in DNA structure upon nucleosome formation, but will require further study. This pattern of shifting indicates a greater affinity and/or specificity for DNA upon inclusion of the AT-hook. EMSAs performed with BRG1 GST-tagged AT-hook (Fig. 4a, right) result in a shifting pattern similar to the BRD alone, revealing that both the AT-hook and BRD are necessary for the robust interaction seen with the dual construct.

The structure of AT-BRD was probed by NMR spectroscopy. $^1$H-$^{15}$N HSQC spectra of hBRM and BRG1 AT-BRD reveal that resonances corresponding to the AT-hook have largely degenerate chemical shift values in the $^1$H dimension and a greater intensity as compared to BRD resonances (Supplementary Figs 3b and 9a), indicating that the AT-hook is unstructured and dynamic. Comparison of $^1$H-$^{15}$N HSQC spectra for AT-BRD and BRD shows that the presence of the AT-hook does not alter the overall fold of the BRD, as negligible CSPs are seen in the majority of BRD resonances (Supplementary Fig. 3b).The only significant perturbations are localized to the N-terminal residues

of the BRD that flank the AT-hook and in the C-terminal portion of αA and the AB-loop (Supplementary Fig. 3c). The latter could indicate transient interactions of the AT-hook with the AB-loop or changes in the conformation of the AB-loop upon inclusion of the AT-hook. $^1$H-$^{15}$N HSQC titrations of acetylated histone peptides into the hBRM AT-BRD reveal identical binding interfaces (Supplementary Fig. 1c) and dissociation constants as compared to the BRD alone (Supplementary Table 1), demonstrating that the presence of the AT-hook does not alter histone-binding activity.

To further investigate the DNA-binding activity, $^1$H-$^{15}$N HSQC spectra were collected on $^{15}$N-labelled AT-BRD upon titration of DNA. Addition of DNAII led to significant CSPs in both the AT-hook and BRD (Fig. 4b,c). Although the overall magnitude of CSPs was larger, the BRD-binding pocket was nearly identical to that seen for the BRD alone with the exception of the AB-loop (compare Figs 3d and 4c). This suggests that the presence of the AT-hook does not significantly alter the mode of DNA binding by the BRD, but instead makes additional,

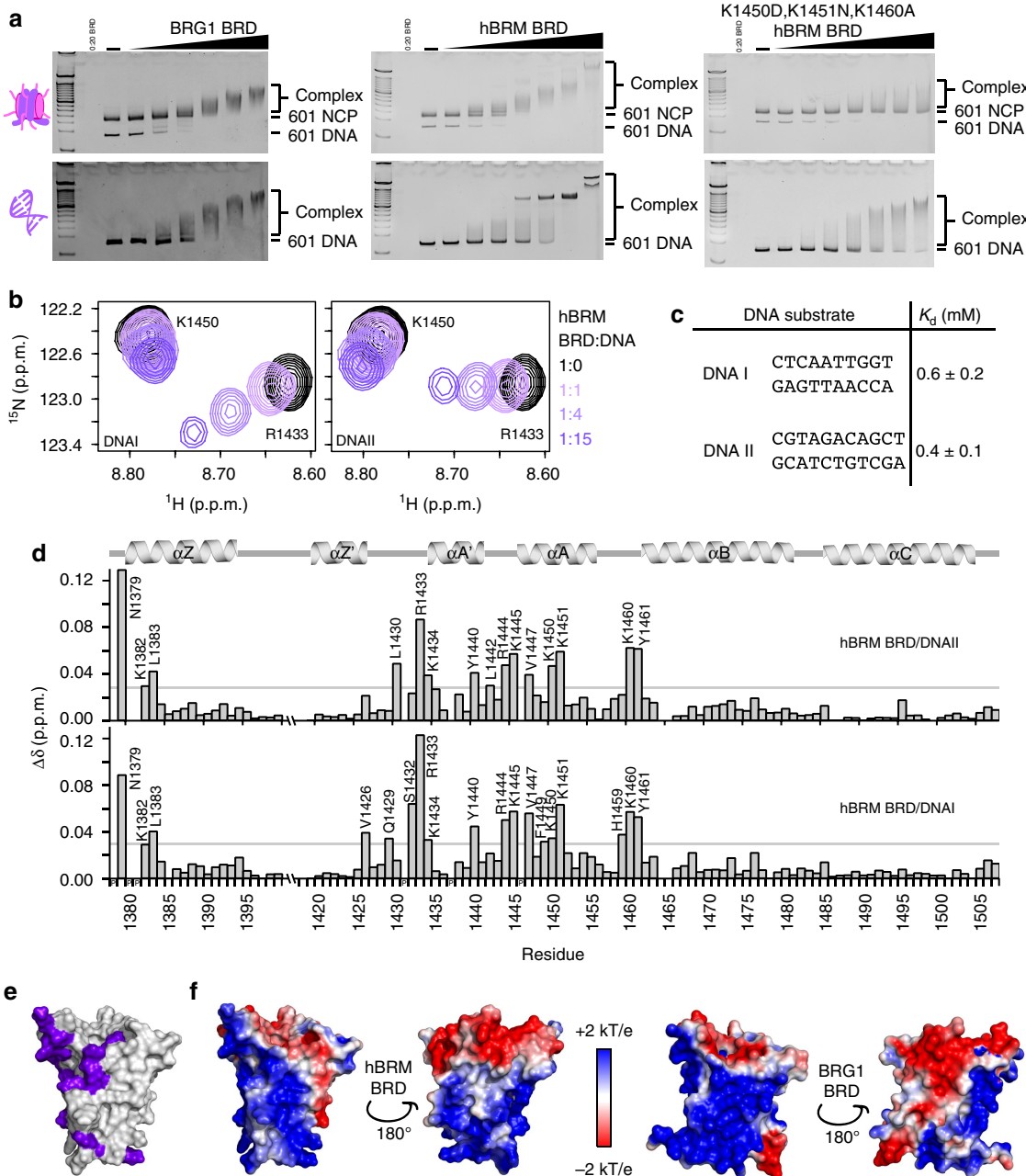

**Figure 3 | The hBRM and BRG1 BRDs interact with linear and nucleosomal DNA through a surface basic patch.** (**a**) EMSAs carried out with BRG1 (left), hBRM (centre), or K1450D,K1451N,K1460A mutant hBRM (right) BRDs and either 601 NCP (top) or free 601 DNA (bottom). Gels were run with a 100 bp DNA ladder in the far left lane and stained with ethidium bromide for visualization. (**b**) Overlay of $^1$H-$^{15}$N HSQC spectra of $^{15}$N-hBRM BRD upon titration of DNAI or DNAII. The selected region displays resonances for R1433 and K1450, and spectra are colour coded according to protein:DNA molar ratio as shown in the legend. DNA titrations were collected at protein:DNA molar ratios of 1:0, 1:0.25, 1:0.5, 1:1, 1:2, 1:4, 1:8 and 1:15 (additionally 1:18 for DNAII). For clarity, only 4 points are displayed. (**c**) Table of dissociation constants determined from NMR titrations with $^{15}$N-hBRM BRD. $K_d$ values are fit to a single-site binding model under ligand-depleted conditions using 8–9 titration points. Shown are the averages over individual residues with significant CSPs (12–19 in total) and associated standard deviation. (**d**) Corresponding normalized CSPs are plotted as a function of residue upon titration of DNAI or DNAII for the protein:DNA molar ratio of 1:15. The secondary structure of the BRD is denoted above the plots, and residues that are perturbed greater than the average plus two standard deviations after trimming off the top 10% of CSPs are labelled. A grey line marks this level of significance for each titration. (**e**) A surface representation of the hBRM BRD structure (PDB ID 2DAT) with residues that are significantly perturbed upon binding to DNAI are coloured in purple. (**f**) APBS-computed surface electrostatics for hBRM and BRG1 BRDs (PDB IDs 2DAT and 2GRC).

independent contacts. CSPs were seen in resonances for the majority of AT-hook residues, with the largest perturbations observed around K1366, just N-terminal to the GRP motif (Fig. 4c). We next tested interaction with DNAI, which contains an AT-rich element (AATT). Titration of DNAI again leads to extensive CSPs in both BRD and AT-hook. Compared to DNAII,

greater CSPs were seen for resonances corresponding to the AT-hook, especially for and around the GRP motif, and select BRD residues (Fig. 4b,c). In addition, G1369 (of the GRP), E1374 and L1376 have distinct chemical shifts as compared to the DNAII-bound state (Supplementary Fig. 6). Although CSPs indicate the same BRD-binding pocket for DNAI and DNAII, in the context

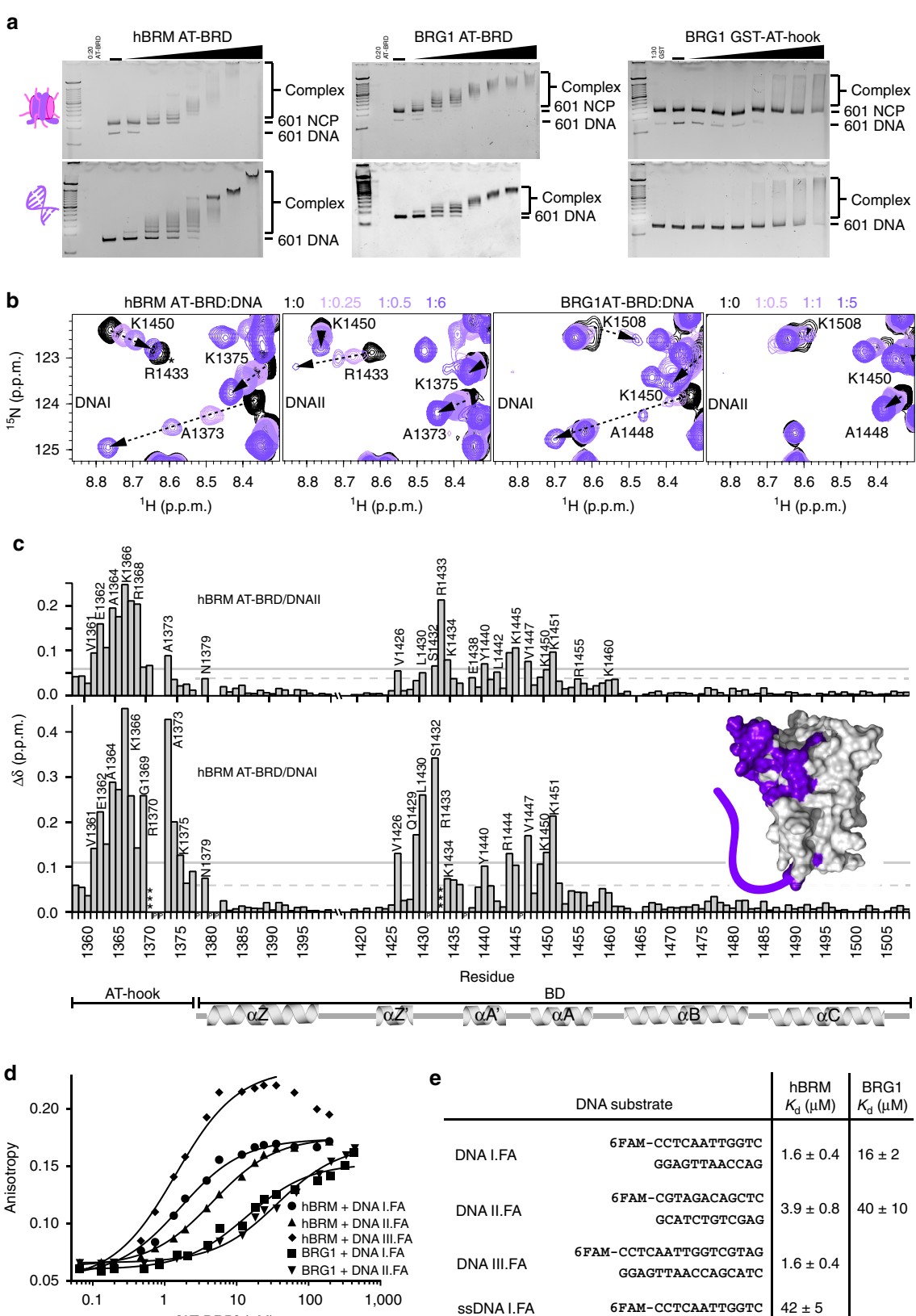

of the AT-BRD the magnitudes of these perturbations are larger upon binding to DNAI. For both DNA titrations, plotting the CSP as a function of DNA concentration indicated stoichiometric binding and thus significantly greater affinity than was observed for BRD alone, consistent with a multivalent mode of association. The same binding activity for DNAI and DNAII was observed in titrations with BRG1 AT-BRD, indicating a nearly identical mode of DNA binding as hBRM (Supplementary Fig. 9b,c).

The affinity of AT-BRD for DNAI and DNAII was determined by fluorescence anisotropy (FA). For these experiments, one strand of the DNA duplexes was labelled with fluorescein, and the DNAs were lengthened by 1–2 bp in order to ensure full duplex formation ($>99\%$) at nanomolar concentrations (referred to as DNAI.FA and DNAII.FA). Titration curves were obtained upon addition of unlabelled AT-BRD (Fig. 4d). hBRM AT-BRD was found to bind DNAII.FA with $K_d = 3.9 \,\mu M \pm 0.8 \,\mu M$ and DNAI.FA with $K_d = 1.6 \,\mu M \pm 0.4 \,\mu M$, demonstrating only modest specificity for the AT-rich DNA (Fig. 4d,e and Supplementary Table 2). It should be noted that other AT-rich sequence variations not tested here may yield higher specificity. Titrations with the longer DNAIII.FA substrate, corresponding to DNAI.FA extended by 4 bp, yields the same affinity as DNAI.FA, suggesting that a single helical turn is likely close to the occluded site size. This agrees with NMR titration data that shows nearly identical binding sites for DNAI and DNAI extended by 4 bp (DNAIII). In addition, plotting the CSP as a function of DNA concentration suggests a 1:1 binding for both substrates (Supplementary Fig. 7). Single stranded DNAI.FA (ssDNAI.FA) yielded $K_d = 42 \,\mu M \pm 5 \,\mu M$, demonstrating that AT-BRD preferentially recognizes double-stranded substrates. BRG1 AT-BRD binds slightly weaker, but still with low micromolar affinity and identical specificity ($K_d = 16 \,\mu M \pm 2 \,\mu M$ and $40 \,\mu M \pm 10 \,\mu M$ for DNAI.FA and DNAII.FA, respectively). Importantly, mutation of the BRD basic patch (see Fig. 3a) decreased the affinity of the AT-BRD for DNA by nearly two orders of magnitude consistent with a true multivalent mode of binding as opposed to an AT-hook driven mode of binding (Supplementary Fig. 8 and Supplementary Table 2).

Together, the data reveal that hBRM and BRG1 AT-hook and BRD make multivalent contacts with DNA, leading to low micromolar affinity with a moderate preference ($\sim 2$-fold) for AT-rich DNA imparted by the AT-hook.

**hBRM BRD can simultaneously bind H3K14ac and DNA.** From CSP mapping, it is evident that there is partial overlap in the DNA and histone-binding pockets of hBRM BRD (Supplementary Fig. 10). Thus, we wanted to test whether the BRD can bind DNA and histone tail simultaneously and whether the two binding events are linked. To investigate this, we titrated

acetylated peptide into $^{15}$N-labelled hBRM AT-BRD pre-bound to DNAI and vice versa. Comparison of $^1$H-$^{15}$N HSQC spectra for BRD or AT-BRD in the (1) apo state, (2) presence of DNA alone, (3) presence of peptide alone or (4) presence of both DNA and peptide provides insight into possible ternary complex formation and communication between binding pockets.

Titration of H3K14ac into hBRM AT-BRD pre-bound to DNAI led to significant CSPs, primarily in resonances in the histone-binding pocket. The observed shifts can be categorized into three main groups. One subset of BRD resonances, which are sensitive to the individual binding of DNA and H3K14ac, move towards CS values that are distinct from the DNAI- or H3K14ac-bound states (Fig. 5a). This demonstrates that in the presence of DNA and H3K14ac a distinct complex is formed, likely a ternary complex, and that these residues are in a unique environment in this complex. Notably, these resonances are broadened, suggestive of a higher molecular weight complex, which is also consistent with a ternary complex. Resonances for another subset of residues do not shift significantly in the presence of DNAI but upon addition of peptide follow linear trajectories to the H3K14ac-bound state (Supplementary Fig. 11a), indicating that these residues adopt the same bound state as with peptide alone. However, compared to titrations with peptide alone, the resonances are again significantly broadened. Finally, resonances for a small subset of BRD residues move along linear trajectories from the DNAI-bound state towards the H3K14ac-bound state, suggesting that though these residues are affected by both DNA and histone binding, they preferentially adopt the histone-bound state in the ternary complex (Supplementary Fig. 11b). The majority of AT-hook residues are not strongly affected by the addition of H3K14ac, although there is slight movement back towards the apo state (see discussion below; Supplementary Fig. 11c). Interestingly, resonances for AT-hook residues A1373 and E1374 adopt a unique chemical shift upon addition of peptide (Supplementary Fig. 11c). These resonances are not sensitive to H3K14ac in the absence of DNAI, suggesting an altered mode of interaction with the DNA in the ternary complex. Altogether, this data reveal that hBRM AT-BRD can bind simultaneously to DNA and H3K14ac. Calculation of the binding affinity of the DNAI-bound AT-BRD for H3K14ac reveals that association with DNA slightly weakens the affinity for the histone tail, although this is likely due to the lower effective concentration of free histone tail due to interaction with DNA (see discussion below, Fig. 5c and Supplementary Table 1).

Similarly, DNA binding does not increase the affinity of hBRM BRD for a weaker-binding peptide substrate. Rather, we observed that titration of H3K27ac into DNA-bound AT-BRD caused resonances for both the BRD and AT-hook to return towards the apo state along the DNA-binding trajectories, indicating release

**Figure 4 | The hBRM and BRG1 AT-hook and BRD form multivalent interactions with DNA.** (**a**) EMSAs carried out with the hBRM (left) or BRG1 (centre) AT-BRD or BRG1 GST-AT-hook (right) and either 601 NCP (top) or free 601 DNA (bottom). Gels were run with a 100 bp DNA ladder on the left and stained with ethidium bromide for visualization. (**b**) Overlay of $^1$H-$^{15}$N HSQC spectra for $^{15}$N-hBRM (left two panels) or $^{15}$N-BRG1 (right two panels) AT-BRD upon titration of DNAI or DNAII. The spectra are colour coded for protein:DNA molar ratio as shown in the legend. DNA titrations were collected at protein:DNA molar ratios of 1:0, 1:0.25, 1:0.5, 1:1, 1:1.5, 1:3, and 1:5-6. For clarity, only four points are displayed. The (*) marking indicates that the peak broadens beyond detection upon addition of DNA, and thus its trajectory cannot be followed. (**c**) Corresponding normalized CSPs for hBRM-B are plotted as a function of residue upon titration of DNAI or DNAII for the protein:DNA molar ratio of 1:5-6. Residues that broaden beyond detection upon addition of DNA are marked (***). The secondary structure of the BRD is denoted below the plots, and residues that are perturbed greater than the average plus two standard deviations after trimming off the top 10% of CSPs are labelled. A grey line marks this level of significance for each titration. A dashed grey line marks the level of significance when only BRD residues are used to calculate the average. (inset) Surface representation of hBRM BRD structure (PDB ID 2DAT) with residues that are significantly perturbed upon binding to DNAI coloured in purple. The AT-hook is drawn in for illustrative purposes. (**d**) Representative FA binding curves for the hBRM-B and BRG1 AT-BRD upon binding to fluorescein-labelled DNAI.FA, DNAII.FA, and DNAIII.FA. The final three data points were not included in the fit for DNAIII.FA because they are suggestive of a secondary process beyond simple one-site binding. (**e**) Table of dissociation constants for the hBRM and BRG1 AT-BRD binding to fluorescein-labelled DNAs, determined via FA. Shown are the average values over 3-4 replicates and the associated standard deviation.

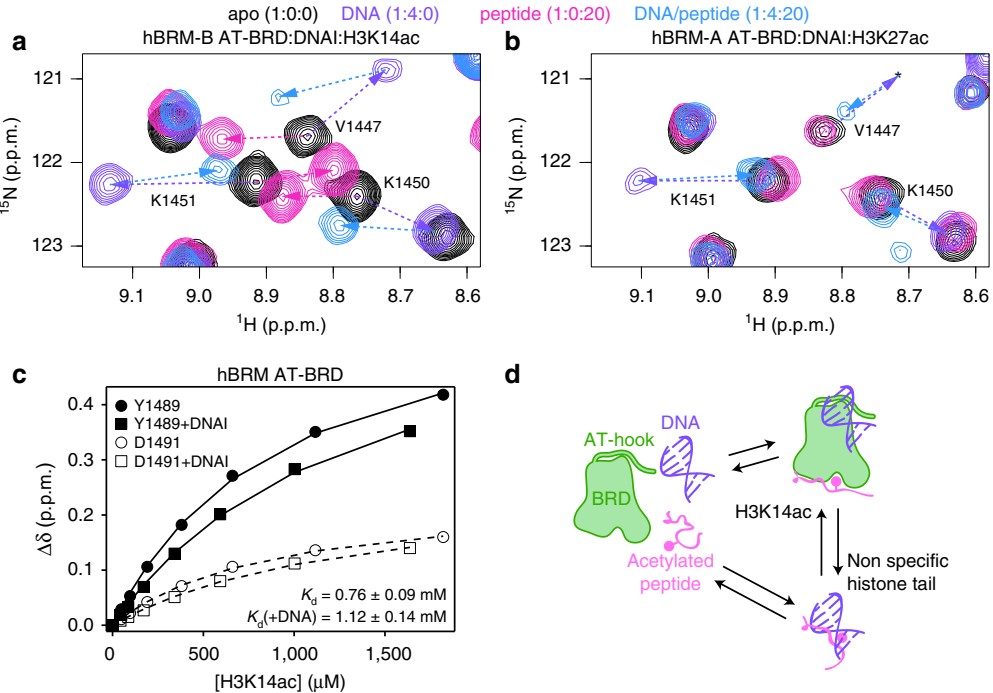

**Figure 5 | The hBRM BRD can bind DNA and H3K14ac simultaneously.** (**a**) Overlay of $^1$H-$^{15}$N HSQC spectra of the $^{15}$N-hBRM AT-BRD apo (black) and in the presence of H3K14ac (pink), DNAI (purple), or both DNAI and H3K14ac (blue). Resonances for V1447, K1450, and K1451 are part of a subset of residues that titrate toward CS values that are distinct from the DNAI- or H3K14ac-bound states. Dashed arrows trace the linear trajectory between apo and H3K14ac-bound (pink), apo and DNAI-bound (purple) and DNAI-bound and both DNAI- and H3K14ac-bound AT-BRD (blue). (**b**) Overlay of $^1$H-$^{15}$N HSQC spectra of the $^{15}$N-hBRM-A AT-BRD apo (black) and in the presence of H3K27ac (pink), DNAI (purple), or both DNAI and H3K27ac (blue). Dashed arrows trace the linear trajectory between apo and DNAI-bound AT-BRD (purple) and upon titration of H3K27ac into DNAI-bound AT-BRD (blue). (**c**) Representative binding curves for Y1489 (closed symbols) and D1491 (open symbols) of the hBRM-B AT-BRD alone (circles) or pre-bound to DNAI (squares) as a function of H3K14ac concentration. (**d**) A model for the equilibrium between the hBRM AT-BRD, acetylated histone peptide, and DNA. Interaction with H3K14ac favors formation of a ternary complex with direct interactions with both DNA and acetylated histone tail while weak binding histones favour a binary complex between the histone tail and DNA.

of DNA from both AT-hook and BRD (Fig. 5b and Supplementary Fig. 12a). This suggests that histone peptide can compete off DNA through direct interaction, which is consistent with other recent studies that have shown strong interactions between histone tails and DNA[37,38]. Addition of H3(5–23) to DNA-bound AT-BRD led to resonances similarly returning towards the apo state (Supplementary Fig. 12b), demonstrating that formation of the ternary complex is indeed dependent on specific interaction with H3K14ac. Together, these titrations reveal an equilibrium between the free species, the ternary complex, and the DNA-peptide binary complex. Unmodified H3 tail and acetylated peptides that bind hBRM/BRG1 BRD non-specifically will preferentially bind to DNA rather than the BRD, whereas H3K14ac will lead to stabilization of a ternary complex (Fig. 5d). Although this binding mechanism remains to be tested in the context of the nucleosome, it strongly suggests a complex mechanism of histone tail binding that should take into account competitive interaction with the DNA.

**The BRD does not drive global BRG1 chromatin association.** To determine whether H3K14ac binding increases the targeting or retention of BRG1/hBRM AT-BRD to nucleosomes, EMSAs were carried out with NCPs containing acetylation at H3K14. Importantly, this modification was specifically installed at H3K14 via Amber suppression methods and is thus present at highly homogenous levels. The shifting pattern observed for the H3K14ac-NCP with hBRM/BRG1 AT-BRD is virtually unchanged as compared to the pattern observed with unmodified NCP

(Figs 4a and 6a and Supplementary Fig. 4a,b), suggesting that recognition of H3K14ac does not increase the affinity for nucleosomes. However, to avoid a direct comparison between different NCP samples, the interaction with the H3K14ac-NCP was further tested using either wild-type BRG1 AT-BRD or an N1540A mutant. Mutation of this Asn retains the fold of the BRD but completely abrogates H3K14ac-binding activity (Supplementary Fig. 9d,e). Assays with either wild-type or N1540A BRG1 AT-BRD show a nearly identical band shift pattern for H3K14ac-NCP (Fig. 6a). Together, this suggests that although BRG1/hBRM BRDs have specificity for H3K14ac, recognition of this acetyl mark does not contribute substantially to nucleosome targeting or retention, which is instead largely driven by contacts with DNA.

We further investigated the BRD histone and DNA binding functions in a cellular context. BRG1 chromatin binding was assessed in mouse ESC lines containing either tagged wild-type BRG1 or the Asn mutation (N1507A in mouse) within the BRD, introduced using homologous recombination methods. Importantly, the human and mouse BRG1 BRD sequences are identical, and this mutation does not alter BAF complex stability, as shown by urea denaturation of immunoprecipitated endogenous BAF complexes (Supplementary Fig. 13). Mouse ESC nuclei from the wild-type and mutant BRD lines were treated with increasing salt concentrations to extract chromatin-associated proteins, revealing that wild-type and mutant BRG1 are retained on chromatin to the same extent (Fig. 6b). This demonstrates that disruption of the acetyl-lysine-binding activity of BRG1 BRD does not alter the global chromatin affinity of BRG1. Similarly, disruption of

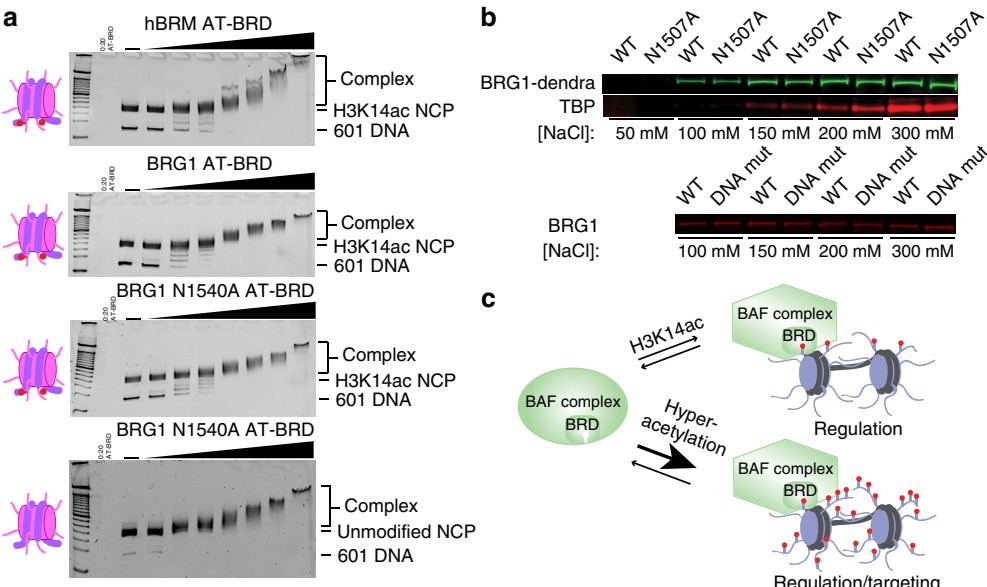

**Figure 6 | Association with H3K14ac does not contribute to nucleosome association.** (**a**) EMSAs carried out with the hBRM AT-BRD or the BRG1 wild-type or N1540A mutant AT-BRD with either the unmodified NCP or the H3K14ac-NCP. Gels were run with a 100 bp DNA ladder in the far left lane and stained with ethidium bromide for visualization. (**b**, top) Differential salt extraction of chromatin-associated proteins from wild-type BRG1-Dendra2 heterozygous mouse ESCs (WT) and BRG1(N1507A)-Dendra2 heterozygous mouse ESCs (N1507A). Wild-type BRG1 and BRG1(N1507A) were detected via blotting of the C-terminal Dendra2 protein tag, with TBP detection serving as loading control. (bottom) Differential salt extraction of chromatin-associated proteins from parental TC1 ESCs (WT) and BRG1 K1508D,K1509N,K1518A-GFP homozygous mouse ESCs (DNAmut). The blot displays BRG1 contained in the free nuclear-supernatant fraction after salt extraction, detected by the G7 anti-BRG1 antibody. (**c**) Proposed model for the role of BRG1/hBRM BRD in the context of the BAF complex in response to acetylation. Upon specific recognition of H3K14ac, BRG1/hBRM BRD causes a change in the BAF complex that regulatesBAF function. Under conditions of hyper-acetylation, the high concentration of even weak binding sites could mean that the BRG1/hBRM BRD additionally plays a role in targeting or retention.

the BRD DNA-binding site does not affect BRG1 affinity for chromatin (Fig. 6b). Together this reveals that the BRD is not critical for global chromatin affinity of BRG1.

## Discussion

In this study, we show that hBRM BRD binds acetylated histone tails weakly, with moderate specificity for H3K14ac, identical to BRG1 BRD. We further show that recognition of H3K14ac by BRG1/hBRM BRD does not substantially contribute to nucleosome targeting. Instead, BRG1/hBRM BRDs target nucleosomes through contacts with DNA. DNA binding appears to be a gain of function in the BRD of SWI/SNF ATPases in higher eukaryotes (Supplementary Fig. 15). Notably, several point and truncation mutations have been identified in human cancers in the DNA binding region of BRG1 and hBRM BRDs (see Supplementary Fig. 15) indicating that this activity is important for the function of BRG1/hBRM and may contribute to tumour suppression[39].

Although well characterized as histone-binding domains, DNA binding is a previously unrecognized property of BRDs. Analysis of the whole class of BRDs suggests that DNA binding ability is not unique to BRG1/hBRM, and in fact we would predict that roughly a third of identified human BRDs may have DNA-binding ability as assessed by their predicted isoelectric point (pI) (Supplementary Fig. 14). In fact while this manuscript was under review one of these predicted DNA binders, the first BRD domain of BRDT, was reported to associate with DNA[40] using a similar basic binding patch. A small subset of other histone effector domains have been reported to bind both histone tails and DNA (or RNA), including some chromodomains, Tudor domains and PWWP domains[41–47]. The functional importance of these DNA contacts in the context of the parent protein or complex is not yet clear for many of these effector domains. For those that bind to modifications close to the nucleosome core, it may be important

to make favourable interactions with DNA in order to stably associate with their cognate histone ligand. However, we and others have recently observed that histone tails can robustly interact with DNA directly[37,38], and many studies have suggested that in the context of the nucleosome the histone tails are robustly associated with the nucleosomal and/or linker DNA (for review see ref. 48). This suggests that histone tail accessibility is likely to be a factor in effector domain binding and should be considered. It is possible that the DNA-binding ability of some of these domains not only helps to stabilize them at the nucleosome but may be important in displacing the histone tails from DNA. Further studies are needed to determine the *in vivo* functional importance of the AT-BRD DNA-binding activity. It is possible that it may contribute to chromatin targeting and/or retention, facilitated diffusion along chromatin, positioning of the BAF complex on nucleosomes, or histone tail displacement.

Our observation that acetyl-lysine recognition does not substantially contribute to *in vitro* nucleosome targeting or chromatin affinity in mouse ESCs is in agreement with recent studies using inhibitors of hBRM/BRG1 BRD. These studies demonstrate that under normal cellular levels of histone acetylation, inhibition of hBRM/BRG1 BRD acetyl-lysine binding does not lead to a measureable decrease in recovery time in FRAP assays[17,18,25]. However, treatment of ESCs with the BRD inhibitor PFI-3 did lead to defects in stem cell maintenance and differentiation, suggesting that acetyl-lysine binding is in fact critical for proper BAF function[17]. Interestingly, if cells were pre-treated with HDAC inhibitor, increased recovery times were observed that were lessened upon treatment with inhibitor[17,18,25]. This suggests that the BRD may be important in retention at hyper-acetylated nucleosomes and is consistent with other studies that have demonstrated that the BRD is only measurably important for the targeting of BRG1 to chromatin under

conditions of hyper-acetylation[17–20,25]. This is consistent with our results using a multi-acetylated histone tail, for which we saw increased affinity.

Together, our results are consistent with a model in which the BRD plays a regulatory role in BRG1/hBRM function in response to H3K14ac, while perhaps playing a role in targeting or retention under conditions of hyper-acetylation (Fig. 6c). Regulation by histone acetylation will be even further modulated through incorporation of other BRD-containing subunits such as BAF180. There are now several examples of mechanisms in which the recognition of the cognate histone ligand by a histone effector domain regulates the host protein or complex beyond straightforward targeting, acting to increase activity through allosteric activation or a conformational change that releases auto-inhibition (for example, see refs 49–52). In fact, the yeast homologue of BRG1/hBRM, Snf2, has itself been shown to be acetylated, and intramolecular recognition by its own BRD negatively regulates chromatin remodelling activity[53,54]. Although these Lys residues are not conserved in BRG1 or hBRM, both are known to be acetylated at other residues *in vivo*[55,56]. Further studies are needed to determine how recognition of H3K14ac by BRG1/hBRM BRD may regulate the function of the BAF complex. Acetylation of H3K14 is important in the DNA damage response pathway[57]. In addition, this mark is found at the promoters of active genes and is correlated with gene expression, and has also recently been found to be associated with the promoters of inactive inducible genes in mouse ESCs[58,59]. Interestingly, recent studies in mouse ESCs indicate that BRG1 has a critical role in regulating the level of H3K14ac at promoters, though it is not yet clear if the BRD has any function in this pathway[60].

Altogether, these studies suggest that BRG1/BRM BRD has evolved to respond to a dynamic chromatin environment, especially to provide a mechanism for functional response to varying levels of histone acetylation. In addition, they reveal a previously unrecognized function of the BRD in its ability to associate with linear and nucleosomal DNA and the ability to form multivalent interactions with DNA in tandem with the AT-hook. These studies should be useful in informing the development of inhibitors of BRG1/hBRM BRDs for use in BAF complex inhibition.

## Methods

**hBRM and BRG1 constructs.** The codon optimized hBRM-A AT-BRD gene fragment (residues 1358-1508) was obtained from Integrated DNA Technologies (IDT) and cloned into the pDEST15 vector using Invitrogen Gateway recombination cloning technology (ThermoFisher Scientific). A PreScission Protease cleavage site was engineered onto the N-terminal side of the sequence. hBRM-A BRD (residues 1379–1508), hBRM-B AT-BRD (residues 1358–1399 and 1418–1508) and hBRM-B BRD (residues 1379–1399 and 1418–1508) constructs were generated from this fragment using the Q5 mutagenesis kit (New England Biolabs). The hBRM K1450D,K1451N,K1460A BRD mutant was generated using the Q5 mutagenesis kit (New England Biolabs). The BRG1 AT-hook (residues 1434-1454), BRD (residues 1454-1569) and AT-BRD (AT-BRD) (residues1434–1569) of the human BRG1 subunit were subcloned out of full-length BRG1 obtained from GE Open Biosystems and cloned into pDEST15 and pGEX6p1 vectors.

One Shot BL21 (DE3) or Rosetta2 (DE3) pLysS Chemically Competent *Escherichia coli* (ThermoFisher Scientific) were used to express all constructs. *E. coli* was grown in LB media or M9 minimal media supplemented with vitamin (Centrum), $1\,\mathrm{g}\,\mathrm{l}^{-1}$ $^{15}$NH$_4$Cl and either $5\,\mathrm{g}\,\mathrm{l}^{-1}$ D-glucose or $3\,\mathrm{g}\,\mathrm{l}^{-1}$ $^{13}$C D-glucose to produce $^{15}$N- or $^{15}$N,$^{13}$C-isotopically enriched protein. Bacteria was grown to an OD$_{600}$ ∼1.0 and induced with 0.3 mM IPTG at 20 °C (hBRM) or 28 °C (BRG1) for 16–20 h.

For purification of the GST-fusion proteins, cells were lysed via freeze-thaw and sonication in 20 mM Tris pH 7.5, 500 mM NaCl, 3 mM DTT, 0.5% Triton X-100, $0.5\,\mathrm{mg}\,\mathrm{ml}^{-1}$ lysozyme with DNaseI and Pierce Protease Inhibitor Tablets (ThermoFisher Scientific). The soluble portion of the lysate was incubated with glutathione agarose resin (ThermoFisher Scientific) and washed extensively with buffer (50 mM potassium phosphate pH 7.0, 50 mM KCl, 1 mM DTT, 0.5 mM EDTA). Samples were cleaved from the GST tag overnight with PreScission Protease and further purified using cation exchange and size exclusion chromatography (Superdex 75 10/300, GE Healthcare Life Sciences). The final buffer for all samples was 50 mM potassium phosphate pH 7.0, 50 mM KCl, 1 mM DTT and 0.5 mM EDTA.

**Histone purification and NCP reconstitution.** Unmodified human histones were expressed out of pET3a vectors in Rosetta 2 (DE3) pLysS (Novagen). Growths were carried out in either M9 or LB media and were induced at OD$_{600}$ ∼0.4 with 0.2 mM (for H4) or 0.4 mM (for H2A, H2B and H3) IPTG for 3–4 h. Histones were extracted from inclusion bodies following[61] and purified by ion exchange chromatography. Histone H3 specifically acetylated at the K14 position was generated using Amber suppression methods using plasmids obtained from the Chin laboratory. Briefly, H3 was cloned into the pCDF-PylT plasmid containing the amber suppressor tRNA, and the codon corresponding to K14 was mutated to TAG using the Q5 Site Directed Mutagenesis Kit (New England BioLabs). Expression and purification methods were adapted from ref. 62. Briefly, BL21 (DE3) cells were co-transformed with the pCDF-PylT plasmid and the pAcKRS-3 plasmid containing the tRNA synthetase. Transformed cells were grown overnight in LB and induced at OD$_{600}$ ∼0.8. The culture was supplemented with 20 mM nicotinamide and 10 mM acetyl lysine, induced 30 min later with 0.5 mM IPTG, and harvested 4 h after induction. The K14ac-H3 was extracted from inclusion bodies, purified on a Ni$^{2+}$ column, and cleaved from the His-tag by TEV protease. Final purification of the histone was carried out by ion exchange chromatography. Unmodified and H3K14ac- octamers were prepared as described in ref. 61. Briefly, equimolar ratios of histones were combined in 20 mM Tris pH 7.5, 6 M Guanidine HCl, 10 mM DTT, dialysed into 20 mM Tris pH 7.5, 2M KCl, 1 mM EDTA, 5 mM BME and purified over a sephacryl S-200 column by FPLC.

A plasmid containing 32 repeats of the 147 bp Widom 601 sequence was amplified in *E. coli* and purified via alkyline lysis methods essentially as outlined in ref. 61. The 601 repeats were released by cleavage with EcoRV and purified from parent plasmid via polyethylene glycol precipitation.

Unmodified and H3K14ac-NCPs were reconstituted with the 147 bp Widom 601 sequence via desalting methods[61]. Briefly, octamer and 601 DNA were mixed at a 1:1.1–1.2 molar ratio and desalted using a linear gradient from 2 M to 150 mM KCl over ∼48 h. NCPs were heat-shocked at 37 °C for 30 min to obtain uniform positioning and then purified via sucrose gradient.

**DNA samples.** Oligos for use in NMR studies were obtained from IDT with the following sequences: DNAI (5′-CTCAATTGGT-3′ and 5′-ACCAATTGAG-3′), DNAII (5′-CGTAGACAGCT-3′ and 5′-AGCTGTCTACG-3′) and DNAIII (5′-CCTCAATTGGTCGTA G-3′ and 5′-CTACGACCAATTGAGG-3′). Double-stranded oligos were annealed at a concentration of 350 μM by heating to 94 °C for 10 min followed by a slow cooling to room temperature. The annealed duplexes were purified by size exclusion chromatography (Superdex 75 10/300, GE Healthcare Life Sciences) in buffer (50 mM potassium phosphate pH 7.0, 50 mM KCl, with 0.5 mM EDTA in most samples) and concentrated. At the concentrations used for NMR titrations, these short dsDNAs are predicted to be at least 98% duplex.

For FA experiments, HPLC-purified 5′-fluorescein-labelled oligos, along with unlabelled complementary strands, were obtained from IDT and annealed as described above. To ensure that the dsDNA was at least 99% hybridized at the low concentrations used in FA, the DNA sequences used for NMR experiments were lengthened by 1–2 bp such that the DNA sequences are: DNAI.FA (5′6-FAM-CCTCAATTGGTC-3′ and 5′-GACCAATTGAGG-3′), DNAII.FA (5′6-FAM-CGTAGACAGCTC-3′ and 5′-GAGCTGTCTACG-3′) and DNAIII.FA (5′6-FAM-CCTCAATTGGTCGTAG-3′ and 5′-CTACGACCAATTGAGG-3′).

**Peptide samples.** Histone peptides were obtained from AnaSpec (H3(5-23), H3(23-34), H3(9-19)K14ac, H3(23-34)K27ac, H3(1-30)K(14,18,23,27)ac-GGK(biotin) and H4(1-20)K8ac, all at ≥95% purity) or the Protein Facility at the Iowa State University Office of Biotechnology (H3(18-28)K23ac and H4(15-25)K20ac, ∼86% and 80% purity, respectively). Peptide stocks were prepared based on weight in buffer (50 mM potassium phosphate pH 7.0, 50 mM KCl, 1 mM DTT, with 0.5 mM EDTA in most samples). When necessary, the pH of stock solutions was adjusted to ensure pH 6–7.

**NMR spectroscopy and data analysis.** To obtain backbone assignments for the hBRM-A and BRG1 AT-BRD, HNCA (hBRM only), HNCACB and CBCA(CO)NH spectra were collected on a 0.5–0.8 mM $^{15}$N,$^{13}$C-labelled sample using a 600 MHz Varian spectrometer at 20 °C (for hBRM) or 37 °C (for BRG1). Data was processed in NMRPipe[63] and analysed using CcpNmr Analysis[64]. For the HNCA, data were collected with 32 scans and 40 and 32 complex increments in the $^{13}$C- and $^{15}$N-dimensions, respectively. For hBRM, the HNCACB was collected with nonuniform sampling with 80 scans and 25% sampling of 80 and 40 complex increments in the $^{13}$C- and $^{15}$N-dimensions, respectively. The HNCACB was reconstructed using hmsIST[65]. The CBCA(CO)NH was collected with 16 scans and 60 and 32 complex increments in the $^{13}$C- and $^{15}$N-dimensions, respectively. Assignments were compared with Biological Magnetic Resonance Data Bank (BMRB) entry 11329 for hBRM-B (residues 1377-1399 and 1418-1504). For residues with poor signal-to-noise in hBRM-A spectra but with resonances in$^1$H-$^{15}$N HSQC spectra collected on hBRM-B constructs that overlayed well with the BMRB-deposited assignments, assignments were transferred from entry 11329. The assignments for hBRM-B were deduced from h-BRM-A. Uniform sampling methods were used for all experiments on the BRG1 constructs. Assignments for BRG1 at 25 °C were deduced through temperature titration.

Titrations of peptide and DNA were carried out by collecting $^1$H-$^{15}$N HSQC spectra on $^{15}$N hBRM or BRG1 AT-BRD or BRD constructs in the apo state and with increasing concentrations of substrate. Protein samples were at 100 μM in 50 mM potassium phosphate pH 7, 50 mM KCl, 0.5 mM EDTA, 1 mM DTT and 10% D$_2$O. Peptide titrations were collected at protein:peptide ratios of 1:0, 1:0.5, 1:1, 1:2, 1:4, 1:7, 1:12 and 1:19-20. Free acetyl-lysine titrations were collected at protein:Kac ratios of 1:0, 1:0.5, 1:1, 1:2, 1:4, 1:7, 1:12, 1:20 and 1:50. DNA titrations were collected at BRD:DNA ratios of 1:0, 1:0.25, 1:0.5, 1:3, 1:6 and 1:10 or 1:0, 1:0.25, 1:0.5, 1:1, 1:2, 1:4, 1:8, 1:15-16, and sometimes 1:18 and AT-BRD ratios of 1:0, 1:0.25, 1:0.5, 1:0.75-1, 1:1.5, 1:3 and 1:6. Data were collected at 20 and 25 °C for hBRM and BRG1, respectively, on an 800 MHz Bruker spectrometer with a cryo probe or 600 MHz Varian spectrometer. Titration data were processed in NMRPipe[63] and analysed using CcpNmr Analysis[64]. Binding curves were fit using a nonlinear least-squares analysis in Igor (Wavemetrics) or GraphPad Prism 6 to a single-site binding model under ligand-depleted conditions:

$$\Delta\delta = \Delta\delta_{max}\left(([L] + [P] + K_d) - \sqrt{([L] + [P] + K_d)^2 - 4[P][L]}\right)/(2[P]),$$

where [P] is the concentration of protein, [L] is the concentration of histone peptide or DNA, $\Delta\delta_{max}$ is the normalized chemical shift change at saturation. The normalized chemical shift difference ($\Delta\delta$) at each point in the titration is calculated by

$$\Delta\delta = \sqrt{(\Delta\delta_H)^2 + (0.154\Delta\delta_N)^2},$$

where $\Delta\delta_H$ and $\Delta\delta_N$ are the changes in the $^1$H and $^{15}$N chemical shift, respectively, at each titration point with respect to the apo chemical shifts. Reported $K_d$ values were determined by fitting $K_d$ values to individual residues with significant CSPs and calculating the average and standard deviation of the $K_d$ values for these residues. Residues with fit $K_d$ values not within the average ± two standard deviations were removed to obtain the reported mean and standard deviation. Residues were determined to be significantly perturbed if the $\Delta\delta$ was larger than the average plus two standard deviations of the $\Delta\delta$ values for all residues after trimming the largest 10%.

**Fluorescence anisotropy.** FA titrations were carried out using a Horiba Scientific Fluorolog-3 fluorometer running FluorEssence V3.5 software. Temperature was maintained at 20 °C in the sample chamber using a water bath. Fluorescein-labelled samples were excited at 492 nm with emission recorded at 515 nm and 5 nm excitation and emission bandpass. Anisotropy data were collected with 1 s integration time as an average of ten values unless a 2% error tolerance was reached in fewer scans. G factors ($I_{HV}/I_{HH}$) were calculated for each point in the titration and were consistent across titrations. Anisotropy ($r$) is calculated as

$$r = \frac{I_{VV} - GI_{VH}}{I_{VV} + 2GI_{VH}},$$

where $I_{VV}$ is the fluorescence emission intensity measured with vertically polarized excitation and vertically polarized emission, $I_{VH}$ is the intensity measured with vertically polarized excitation and horizontally polarized emission and $I_{HH}$ and $I_{HV}$ are the intensity measured with horizontally polarized excitation and horizontally or vertically polarized emission, respectively.

Titration samples contained 100 nM 5′-fluorescein-labelled DNA in 50 mM potassium phosphate pH 7, 50 mM KCl, 0.5 mM EDTA, 1 mM DTT (129 mM total [K$^+$]) in a 3 mm quartz cuvette with initial volumes of 150 μl. Unlabelled hBRM or BRG1 constructs were titrated into these solutions. Binding curves were fit using nonlinear regression analysis in GraphPad Prism 6 to a simple single-site binding model according to

$$r = r_0 + (r_{max} - r_0)\frac{x}{K_d + x},$$

where $r_0$ is the anisotropy value for the free DNA, $r_{max}$ is the anisotropy value for the DNA-protein complex, $x$ is the free concentration of protein, which is assumed to be equal to the total concentration of protein under the condition of the assay, and $K_d$ is the dissociation constant.

Titrations were collected in triplicate or quadruplicate with binding affinities reported as the average and standard deviation of the fits to each individual titration. DNAI.FA titrations with hBRM-B AT-BRD were collected at both 50 and 100 nM DNA to confirm identical results. For the titration of the hBRM-B AT-BRD into DNAIII.FA, the final three data points were omitted from the fits. The reproducible decrease in FA at high concentrations of protein is suggestive of a secondary process beyond the simple one-site binding of interest for this study that affects the conjugated fluorescein. DNA that is longer than the occluded site size of the protein will lead to the ligand–lattice interactions of nonspecific DNA binding, which is beyond the scope of this publication.

**Electrophoretic mobility shift assays (EMSAs).** EMSAs were run using 5% 75:1 acrylamide:bisacrylamide native gels. Gels were prepared and run in 0.2 × TBE. Each gel sample contained 1.5 pmol of DNA or NCP (150 nM final concentration) with 10 mM potassium phosphate and 10 mM KCl. hBRM or BRG1 constructs were added at DNA or NCP:protein ratios of 0:20, 1:0, 1:1, 1:4, 1:6, 1:10, 1:15, 1:20 and 1:30 (with 1:30 omitted on some gels) run alongside a 100 bp ladder. Samples were incubated on ice for 30 min prior to loading on the gel. Gels were run on ice

for 45–50 min at 125 V and visualized via staining with ethidium bromide followed by imaging using an ImageQuant LAS 4000 imager. Representative uncropped gels can be seen in Supplementary Fig. 16.

**Circular dichroism (CD).** CD scans and thermal denaturation curves were collected on a Jasco J-815 CD spectrometer using a 1 mm quartz cuvette. Protein was at a concentration of 20 μM in 50 mM potassium phosphate pH 7, 50 mM KCl, 0.5 mM EDTA. Three averaged scans were collected across 190–260 nm at temperatures ranging from 5–95 °C in 2 °C increments with a 5 min equilibration at each temperature. Thermal denaturation curves were produced by plotting the raw observed ellipticity at 222 nm as a function of temperature. Data were fit in Igor (Wavemetrics) to the following set of equations based on the Gibbs–Helmholtz equation for unfolding as a function of temperature:

$$\Delta G = \Delta H(1 - T/T_M) - \Delta C_p((T_M - T) + T\ln(T/T_M)),$$

$$K = \exp(-\Delta G/(RT)),$$

$$\alpha = K/(1 + K),$$

$$[\theta]_T = \alpha(([\theta]_F + m_F T) - ([\theta]_U + m_U T)) + ([\theta]_U + m_U T),$$

where $\Delta G$ is the free energy of folding, $\Delta H$ is the enthalpy of folding, T is the temperature recorded for each point, $T_M$ is the temperature at which $\alpha = 0.5$, $\Delta C_p$ is the change in heat capacity between the folded and unfolded states and assumed to be equal to zero, $K$ is the equilibrium constant of folding, $R$ is the gas constant and $\alpha$ is the fraction folded at a given temperature. The ellipticity at 222 nm measured at each temperature, $[\theta]_T$, was fit by including linear corrections to the fully folded ($[\theta]_F$) and fully unfolded ($[\theta]_U$) ellipticities. Unfolding curves as a function of temperature were then simulated using the fit values for $\Delta H$ and $T_M$.

**Modelling of bromodomains.** The phylogenetic tree for human BRDs was constructed using the open-access, online resource ChromoHub[66].

To compare the electrostatics of orthologues of hBRM/BRG1 BRD across the ATPase subunit(s) of the SWI/SNF complex in different organisms, the structures of these orthologue BRDs were modelled onto a known structure. Domain boundaries for each BRD were determined using SMART (a Simple Modular Architecture Research Tool). The structure of each BRD was modelled onto that of the BRG1 BRD (PDB ID: 2GRC) using SWISS-MODEL[67]. Surface electrostatics were computed using the APBS plug-in for PyMOL[68](MG Lerner and HA Carlson, APBS plugin for PyMOL, 2006, University of Michigan, Ann Arbor.) using default parameters (including protein and solvent dielectric constants of 2.0 and 78.0, respectively) and visualized using PyMOL. Theoretical pI values were calculated using ProtParam (ExPASy)[69].

**Mouse embryonic stem cell genetic targeting and culture.** A bacterial artificial chromosome (BAC) containing part of the mouse Brg locus was modified through recombineering to create the BrgN1507A mutation, a Brg C-terminal Dendra2 tag, and a neomycin-resistance gene downstream of the Brg 3′-untranslated region (UTR). The BAC region of interest was captured into a targeting vector. Linearized targeting vector (20 μg) was electroporated into 10e6 early passage wild-type male TC1 ESCs (129/Sv mouse strain) in order to generate heterozygous clones through homologous recombination.

After G418 selection, surviving colonies were picked and screened by PCR for correct insertion of the 3′-end of the targeting vector. A selection of positive clones was further screened by Southern blot for correct insertion and for copy number of the targeting vector. Positive clones were also sequenced through the BRD to confirm that they were heterozygous for the N1507A point mutation. The majority of positive clones had the point mutation, but several had only wild-type BRD sequence, indicating that the 3′-end of the *Brg* locus was targeted with the Dendra2 tag and neo-resistance, but the 5′-end of the targeting vector including the point mutation was not incorporated. One of these lines was used as a wild-type Brg-Dendra2 control in experiments utilizing the heterozygous BrgN1507A-Dendra2 ESC line (Supplementary Fig. 13 and Fig. 6b).

During targeting procedures, ESCs were grown under standard serum/LIF culture conditions on mouse embryonic fibroblast-coated plates: knockout Dulbecco's modified eagle medium supplemented with 15% ES-qualified FBS (Invitrogen), 2 mM L-glutamine (Gibco), 10 mM Hepes (Gibco), 100 U ml$^{-1}$ penicillin/streptomycin (Gibco), 0.1 mM non-essential amino acids (Gibco), 0.1 mM beta-mercaptoethanol (Gibco), and ~1,000 U ml$^{-1}$ LIF (Crabtree Laboratory).

For differential salt extraction and urea denaturation experiments, ESCs were grown on gelatin-coated plates under naive 2i/LIF conditions in N2B27 media according to ref. 70 supplemented with ~1,000 U ml$^{-1}$ LIF (Crabtree Laboratory), 1 μM PD0325901 (Stemcell Technologies) and 3 μM CHIR99021 (Stemgent). All ESCs were maintained at 37 °C, 5% CO$_2$, with daily media replacement, and passaged onto new plates following trypsin dissociation every 48–72 h.

**Urea denaturation of protein complexes.** ESC nuclei were isolated by resuspending cells in hypotonic buffer (10 mM Hepes (pH 7.6), 25 mM KCl, 1 mM EDTA, 10% glycerol, 1 mM DTT, and protease inhibitors) on ice. Nuclei were

sedimented by centrifugation (1,000*g*), resuspended (10 mM Hepes (pH 7.6), 3 mM MgCl2, 100 mM KCl, 0.1 mM EDTA, 10% glycerol, 1 mM DTT and protease inhibitors), and lysed by addition of ammonium sulfate solution (final concentration of 0.3 M). Soluble nuclear proteins were separated from insoluble chromatin by ultracentrifugation (100,000*g*) and then the nuclear proteins were precipitated with 0.3 mg ml$^{-1}$ ammonium sulfate powder for 20 min on ice. Protein precipitate was sedimented by ultracentrifugation (100,000*g*) and then resuspended in IP buffer (150 mM NaCl, 50 mM Tris-HCl (pH 8.0), 1% NonidetP-40, 0.5% deoxycholate, 1 mM DTT, 1 mM PMSF with protease inhibitors) for urea denaturation studies. For urea denaturation analysis, 150 µg aliquots of nuclear extracts resuspended in 25 µL IP buffer were incubated with an equal volume of urea solution for 15 min at room temperature. These solutions were then diluted approximately sevenfold in cold IP buffer with protease inhibitors and used for subsequent immunoprecipitation (IP) of Dendra2. For each IP, 2.5 µg of anti-Dendra2 antibodies were incubated with each 150 µg urea-denatured nuclear extract at 4 °C overnight with rotation. After incubation, 15 µl of protein G Dynabeads were used to isolate Ig-protein complexes, incubating for 2 h with rotation at 4 °C. Ig-protein complexes were washed three times with IP buffer and then boiled in 1.5 × gel-loading dye (NuPage LDS Sample Buffer plus 2% beta-mercaptoethanol). Supernatants were loaded directly onto SDS–polyacrylamide gel electrophoresis gels for western blot analysis (see below).

**CRISPR-Cas9-based cell line construction.** The BRG1 K1508D,K1509N, K1518A-GFP homozygous mutant ESC line was created using CRISPR-Cas9 technology. TC1 ESCs (male line derived from 129/Sv) were targeted using wild-type Cas9 and homology-directed repair (HDR). For HDR, we used a 5 kb segment of *Brg* genomic DNA, including BRD mutations and a C-terminal GFP, inserted into the PUC19 vector backbone. The HDR template was designed so that guide RNAs would not target it due to altered PAM sites. Isolated HDR plasmid (5 ug) was nucleofected into 3 million ESCs along with two Cas9/guide RNA plasmids (2.5 µg each), one targeted to the *Brg*K1508,K1509 and another to the *Brg* 3'UTR. The following partially complementary oligos were annealed and cloned into the pSpCas9(sgRNA) vector to create the two different Cas9/guide RNA plasmids (5' → 3' sequences):

1) CACCGTCAAGAAGATCAAGGTGCGC
AAACCGCGCACCTTGATCTTCTTGA
2) CACCGTTGGCTGGGACGAGCGCCTC
AAACCGAGGCGCTCGTCCCAGCCAA

After nucleofection (according to Amaxa protocols for mouse ESCs, program A013), ESCs were plated at $1 \times 10^6$ per 60 mm dish on antibiotic-resistant irradiated mouse feeder cells. After 24 h, cells were treated with 1.5 µg ml$^{-1}$ puromycin for 48 h. Approximately 7 days after nucleofection, single colonies were picked into individual wells of feeder-coated 24-well plates for maintenance and propagation. Recovered clones were screened by PCR for the C-terminal GFP, and positive clones were further screened by sequencing through the BRG1 BRD to determine zygosity of the point mutations. For putative homozygous clones, a 2 kb PCR product centred on the BRD mutations was fully sequenced with tiled primers in order to ensure accurate repair of the locus and to confirm homozygosity.

**Differential salt extraction of chromatin-associated protein.** Nuclei from ESCs were resuspended in buffer A (25 mM Hepes, pH7.6, 5 mM MgCl2, 25 mM KCl, 0.05 mM EDTA, 10% glycerol, 0.1% NP-40, protease inhibitors) at a concentration of 60 million nuclei per ml. Aliquots of 25 µl were adjusted to the desired NaCl concentration and brought to a final volume of 50 µl. Samples were gently mixed and incubated on ice for 10 min, followed by high-speed centrifugation for 15 min to pellet insoluble chromatin. Nuclear lysate supernatants were removed and mixed with 4 × gel-loading dye for western blot analysis (see below).

**Western blot analysis and antibodies.** For western blot analysis, samples in gel-loading dye were loaded into precast gels (NuPage Bis-Tris 4–12%) and separated by SDS–polyacrylamide gel electrophoresis. Next, they were transferred to polyvinylidene fluoride (PVDF) membranes (Immobilon-FL; Millipore) and blocked with 5% Bovine Serum Albumin for 1 h. Blots were incubated overnight with primary antibodies, washed 3 times in PBS-Tween, then incubated with LiCor infrared secondary antibodies for 1 h (see below for antibody information). After washing in PBS-Tween followed by PBS, blots were visualized on a LI-COR Odyssey CLx. Antibodies used and dilution factors were as follows:

BrgG7, Santa Cruz, sc-17796; WB, used at 1:500.
BAF155, Crabtree Laboratory, used at 1:1,000.
BAF250A, Santa Cruz, sc-32761, used at 1:1,000.
BAF60A, Santa Cruz, sc-135843, used at 1:1,000.
BAF47, Santa Cruz, sc-166165 ×, used at 1:1,000.
Dendra2, Crabtree Laboratory, used at 1:1,000.
TBP, Abcam, ab818, used at 1:3,000.
IRDye 800CW goat Anti-rabbit IgG, LI-COR Biosciences (cat no. 926-32211), used at 1:30,000.
IRDye680 goat Anti-mouse IgG, LI-COR Biosciences (cat no. 926-32220), used at 1:30,000.

**Data availability:** The authors declare that data supporting the findings of this study are available within the paper and its Supplementary Information files, raw data and plasmids are available from the corresponding author upon reasonable request. The assignments for the BRM AT-BRD have been deposited in the Biological Magnetic Resonance Data Bank (BMRB ID 27106).

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

## Acknowledgements

This work was supported in part by the Iowa Cardiovascular Interdisciplinary Research Fellowship (T32HL007121), awarded to E.A.M. E.A.M. is currently supported by an Arnold O. Beckman Postdoctoral Fellowship. J.C.S. is funded by the interdisciplinary Institutional Training Grant in Pharmacological Sciences (T32 GM067795). C.A.M. and work in the Musselman laboratory is funded by an NSF CAREER Award (1452411). J.L.R. was supported in part by an NSF Graduate Research Fellowship. We thank the Carver College of Medicine NMR facility, especially Andrew Fowler for assistance with NMR studies. E.A.M. thanks Drs Greg DeKoster and Chao Wu for advice implementing hmsIST with Varian data. We thank Drs Karolin Luger and Michael Poirier for the gifts of the histone plasmids, Dr Miles Pufall for the gift of the TEV plasmid and Dr Courtney Hodges for the gift of the Dendra2 polyclonal antibody. The plasmids for use in Amber suppression were obtained from the Chin Laboratory.

## Author contributions

Reagents were generated and experiments performed by E.A.M., J.C.S., J.L.R., D.P.F., K.V., J.K.J., B.X.G. and C.A.M. Data were analysed by E.A.M., J.C.S., J.L.R., D.P.F., G.R.C. and C.A.M. E.A.M. and C.A.M. wrote the manuscript with input from J.C.S., J.L.R. and G.R.C.

## Additional information

**Competing interests:** The authors declare no competing financial interests.

**Reprints and permission** information is available online at http://npg.nature.com/ reprintsandpermissions/

