## [Peer Review File · Nature Communications]

Reviewers' comments:

Reviewer #1 (Remarks to the Author):

The paper by Morrison et al "DNA binding drives association of the BRG1 and hBRM bromodomains with nucleosomes indicating a regulatory role for acetyl-lysine binding" reports DNA binding activity of BRG1 and BRM bromodomains through a basic sequence patch formed by the α A helix and the ZA and AB loops. The author's present data suggesting a bidentate binding mode of the BD recognizing Kac histone tails as well as DNA. The DNA interaction has been characterized using NMR titration, FP assays in combination with site directed mutants. The DNA binding properties of BRG1/BRM bromodomains is interesting and due to the conservation of electrostatic surface patches interaction with DNA might be shared by other basic bromodomain proteins. However, the experiments are largely based on in vitro data using isolated bromodomains. Would it not be expected that strong positive surface potential would lead to at least some DNA binding affinity simply due to electrostatic attraction? If this interaction is however physiological relevant remains to be shown.

The SWI/SNF complex constitutes a large multiprotein complex with many DNA and chromatin anchor motifs. The interaction mediated by a single bromodomain is therefore not likely to drastically change chromatin association – which has been demonstrated recently by studies using bromodomain inhibitors. Would it not be more likely that the bromodomain would mediate local interaction which may not necessarily be mediated by histone tails? The particular binding modes of acetyl-lysine suggest that the BRG1/BRM and some of the PB1 bromodomains may also recognize marks that are different from Kac. The weak affinity for its putative recognition sequence studied by the authors (H3K14ac) may make this scenario likely. Nevertheless, the characterization work on the BD DNA interaction is novel and will interest researchers working in the area but importance of the observed interaction with DNA in vitro remains to be clarified. This could be done by expressing mutants that would inhibit DNA association in combination with FRAP or other association assays.

Apart from these general comments I have some specific remarks that the authors should address:

Line 221 following – the discrepancy between published data using ITC and the DNA binding studies presented in the paper are disturbing. Are there differences in buffer conditions and other experimental parameters?

Fig 3F how were the surface electrostatics computed? What is the colour scale shown and under which solvent conditions were the electrostatics computed?

Line 200 following: the insert destabilizes the BD by shifting the unfolding curve $\sim 20^\circ\text{C}$ (Figure 2b). This is puzzling – how can an insert that is unstructured and that does not interact with the BD stabilize the bromodomain by 20 degrees in T_m assays? The low stability indicated that variant A is largely unfolded at 37 degrees. Are both proteins stable and expressed in cells?

Reviewer #2 (Remarks to the Author):

The authors of the manuscript probe the molecular mechanism of BRG1 and BRM bromodomain histone interactions. Histone-effector domain interaction studies have begun to be recognized by the field as an insufficient model for completely capturing relevant nucleosome interactions. As such, more physiologically relevant studies are necessary for rationalizing cellular biology experiments. The authors nicely address this challenge by assessing both histone-bromodomain

interactions as well as DNA and nucleosomal bromodomain interactions using a variety of biophysical methods, including NMR, FP, and EMSA assays and in-cell experiments. Through NMR analysis of various acetylated histone binding interactions, the authors identify a particular acetylation state, H3K14Ac, responsible for bromodomain recognition, albeit weakly. Double stranded DNA was also shown to form an interaction with the bromodomain, as well as a fused AT-hook domain. Using 2D-NMR analysis, the authors provide compelling data to show that a ternary but non-cooperative complex forms, and propose a model that non-specific histone-DNA interactions, can compete with chromatin binding by this effector domain. The experiments were well done, quantitatively analyzed, and provide compelling evidence for the molecular mechanism the authors propose. This research should continue to push the field for studying bromodomain interactions, under more physiological conditions. Although the manuscript is deemed suitable for publications, several (mostly minor) points are raised to be addressed.

1) For Figure 1D the authors note the most significant chemical shift were observed for H3K14Ac. From the text it was unclear what the level of significance was, although in the caption >2 standard deviations was noted as significant. A common convention is to add a 2 standard deviation line to the CSP graph for clarity which would help in this case. From the figure however, it was unclear how many data points were used to fit the K_d . Was it only 4 as indicated in 1f? This seems like this would introduce significant, error, particularly with such weak binders.

2) The authors note multiple times the importance of hyperacetylated states being important for chromatin binding. However, only singly acetylated histones were studied here. BET bromodomains bind to both dual and tetraacetylated histones with the highest affinity. Could a multiply acetylated histone be a more relevant substrate?

3) In Figure 3, the NMR resonance intensity is both shifting and broadening into baseline. From the data it is unclear, if the protein is precipitating out, or if this is reporting on specific protein dynamics. A full spectrum in the SI would help elucidate if this is a residue specific effect or if the protein is crashing out.

4) On page 4 Line 248, the authors note there is about a 2-fold higher affinity for DNA than histones. However, based on the error of the measurements they seem to be close to within error. I'm not sure if this last statement is really necessary.

5) Page 7 Line 301. If the data was good enough to obtain dissociation constants for supplementary table 1, it is unclear why the data is not shown. This is similar to comments on line 323.

6) Page 7 Line 318, rather than a unique trajectory, I think the authors may mean single trajectory which would indicate a 1 to 1 binding event and is consistent with the arrows drawn in the figures. Also the resonance for G1369 and E1374 don't appear in Figure 4c. Are A1373 and K1375 the implied resonances or is another figure being referenced?

7) Line 325 page 8, The pattern of CSPs doesn't necessarily indicate a greater affinity, however, it does support a multivalent interaction.

8) For figure 4e, the FP traces for DNAIII have a much higher anisotropy and what looks like a steeper hill slope. The large increase in anisotropy doesn't seem to be consistent with simply the added nucleotides. Could two bromodomains be interacting with the DNA in this case? The HSQC NMR data with DNAIII might be able to assess that, however the data is indicated again as not shown.

9) The model in Figure 5c nicely captures the data, however equilibrium arrows would be more appropriate than double headed arrows.

10) For labs hoping to use the Amber suppression method it would be helpful to include in the supporting information, the yield and characterization data for the novel construct. For example, is there a lot of truncated product or read through product that might co-purify.

Stylistic/edits

BD was found to be confusing throughout the manuscript due to its similarity with "binding domain" a suggestion would be to write out bromodomain or use Brd.

On line 384, the figure should be 8c and not 6c

We thank the reviewers for their thorough and thoughtful reading of our manuscript and their excellent suggestions. We believe we have been able to address all comments/suggestions, and feel that the revised manuscript is significantly stronger as a result. Please see our response to specific comments below.

Changes other than in response to reviewers' comments include:

- The manuscript has been shorted in some sections to meet word limits of the journal.
- We have added additional data regarding mutational analysis of the AT-BRD (Supplementary Fig. 8 and Table 2) to support the multivalent mode of binding.
- While our manuscript was being revised, a report was published in this journal of another bromodomain from BRDT with DNA binding capability. Indeed, this is one of the bromodomains we predicted would bind to DNA. This important piece of work has of course been recognized and briefly discussed in the revised text.

Reviewer 1

Comment: “(the) importance of the observed interaction with DNA in vitro remains to be clarified. This could be done by expressing mutants that would inhibit DNA association in combination with FRAP or other association assays.”

- Response: We appreciate this point. As suggested we carried out a chromatin association assay with mutant BRD. Specifically, mutant ESC lines were generated using CRISPR technology in which the BRD contains a triple mutation at the DNA binding interface (the same as shown in Figure 3a). Chromatin association of the mutant BAF complexes were assayed by differential salt extraction of chromatin purified from the mutant ESCs, as well as from the mutant cells after differentiation into progenitor neurons. The results of these experiments revealed that that the BRD DNA binding is not playing a major role in global chromatin association of the BAF complex. This is not necessarily surprising given that the ~1MDa BAF complex contains other domains with DNA binding capability, and thus the BRD may not be a major contributor to this particular function. However, the conservation of the DNA binding pocket and extensive mutation of this region in cancer attest to its importance in function. Global chromatin association tested here is only one possible function of the BRD DNA binding (as outlined in the discussion), which could include facilitated diffusion along chromatin, relative occupancy, positioning of the BAF complex on nucleosomes, or histone tail displacement. In order to fully examine the in vivo function it will be necessary to carry out a number of biochemical and functional assays with this mutant as well as BAF constructs containing mutations in both the AT-hook and BRD that are beyond the scope of this current manuscript.

Comment: “the discrepancy between published data using ITC and the DNA binding studies presented in the paper are disturbing. Are there differences in buffer conditions and other experimental parameters?”

- Response: We were certainly surprised by this as well. Buffers are very similar between studies. However, our purification scheme for the BRD was much more extensive including ion exchange chromatography which separates the BRD from a significant amount of nucleic acid contamination from the purification out of *E. Coli*. If this step is omitted a very high 260/280 UV ratio is observed indicating that size exclusion is not sufficient to reach a pure sample. This was not reported in the previous study and thus it is possible that ITC was performed on a “dirty” sample of BRD, contaminated with a significant amount of nucleic acid. In addition, the concentrations used in the ITC experiments for the BRD alone likely precluded the ability to detect DNA binding based on the affinities that we measured by NMR. As this is speculation of another laboratory’s data we would prefer not to comment further in the text itself.

Comment: “How were the surface electrostatics computed? What is the colour scale shown and under which solvent conditions were the electrostatics computed?”

- Response: The surface electrostatics were computed using APBS (Adaptive Poisson-Boltzmann Solver), which has a plug-in for PyMOL. With this program, PDB2PQR prepares the structure for continuum electrostatics calculations. A dielectric constant of 2.0 and solvent dielectric constant of 78.0 were used. The color scale ranges between -2 kT/e and +2 kT/e. To ensure clarity on these points the methods section has been expanded to include these values and the color scale is now labeled in Fig. 3 and Supplementary Fig. 12.

Comment: “the insert destabilizes the BD by shifting the unfolding curve $\sim 20^\circ\text{C}$ (Figure 2b). This is puzzling – how can an insert that is unstructured and that does not interact with the BD stabilize the bromodomain by 20 degrees in T_m assays? The low stability indicated that variant A is largely unfolded at 37 degrees. Are both proteins stable and expressed in cells?”

- Response: Both proteins were expressed identically, which includes expression at 20C, and a GST tag to aid in stability. Though there is no evidence of an interaction between the BD and the insert, internal loops have previously been shown to destabilize protein fold. For example, Nagi and Regan showed that increasing loop length destabilized a four-helix-bundle protein (a similar fold to the bromodomain) (see Nagi and Regan, An inverse correlation between loop length and stability in a four-helix-bundle protein, *Fold Des.* **2**, 67-75 (1997)).

Reviewer 2

Comment: “From the text it was unclear what the level of significance was...A common convention is to add a standard deviation line to the CSP graph for clarity which would help in this case.....how many data points were used to fit the Kd.”

- Reviewer: To clarify both points we have made changes to the figures as well as the text. A standard deviation line has been added to all CSP graphs (Figures 1, 3, and 4). Regarding the calculation of the Kd, 8-9 points were used. This has been clarified in the text and figure legends.

Comment: “Only singly acetylated histones were studied here. Could a multiply acetylated histone be a more relevant substrate?”

- Response: This is an excellent point. To address this, we have carried out an NMR titration with a tetra-acetylated histone peptide. Association with the tetra-acetylated peptide is modestly tighter as compared to H3K14ac, however the bound state is exactly the same. This reveals that the binding pocket can only accommodate a single acetyl mark, and the increase in affinity is likely only due to increased concentration of non-specific sites on the peptide. This additional data is shown in Figure 1 and discussed in the text accordingly.

Comment: “In Figure 3, the NMR resonance intensity is both shifting and broadening into baseline....A full spectrum in the SI would help elucidate if this is a residue specific effect or if the protein is crashing out”

- Response: This is an excellent point. We have now included the full spectra titrations in Supplementary Fig. 5. The line broadening is likely due to the increased size of the complex as well as resonance specific changes in conformational dynamics as alluded to by the reviewer.

Comment: “The authors note there is about a 2-fold higher affinity for DNA than histones. However, based on the error of measurements they seem to be close to within error. I’m not sure if this last statement is really necessary.”

- Response: We agree and have removed this comment from the text.

Comment: “If the data was good enough to obtain dissociation constants for supplementary table 1, it is unclear why the data is not shown.”

- Response: We have added Supplementary Fig. 1c and Supplementary Fig 6b,c with this data.

Comment: “rather than a unique trajectory, I think the authors may mean single trajectory which would indicate a 1 to 1 binding event and is consistent with the arrows drawn in the

figures. Also the resonance for G1369 and E1374 don't appear in Figure 4c. Are A1373 and K1375 the implied resonances or is another figure being referenced?"

- Response: We apologize for the confusion. We are indicating that upon addition of DNAI these resonances follow a trajectory to a bound state that is unique from that observed for DNAII. We have clarified the text and have added Supplementary Fig. 3e showing this data to avoid confusion.

Comment: "The pattern of CSPs doesn't necessarily indicate a greater affinity, however, it does support a multivalent interaction."

- Response: We apologize for the confusion. We have changed the statement to better reflect our intending meaning to "For both DNA titrations, plotting the CSP as a function of DNA concentration indicated stoichiometric binding and thus significantly greater affinity than was observed for BRD alone, consistent with a multivalent mode of association"

Comment: "For figure 4e, the FP traces for DNAIII have a much higher anisotropy and what looks like a steeper hill slope. The large increase in anisotropy doesn't seem to be consistent with simply the added nucleotides. Could two bromodomains be interacting with the DNA in this case? The HSQC NMR data with DNAIII might be able to assess that, however the data is indicated again as not shown."

- Response: In our experience, raw anisotropy values are difficult to predict. A cooperativity factor was not needed to fit the data as shown in Fig. 4d. When plotted as 'fraction bound' rather than 'anisotropy' on the y-axis, the fit curves for DNAI.FA and DNAIII.FA nearly overlay, as expected from the nearly identical fit K_d values. In addition, NMR data indicates that this association is still 1:1 as shown in new Supplementary Figure 7. However, as acknowledged in the text "The reproducible decrease in fluorescence anisotropy at high concentrations of protein is suggestive of a secondary process beyond the simple one-site binding of interest for this study that affects the conjugated fluorescein. DNA that is longer than the occluded site size of the protein will lead to the ligand-lattice interactions of nonspecific DNA binding, which is beyond the scope of this publication."

Comment: "The model in Figure 5c nicely captures the data, however equilibrium arrows would be more appropriate than double headed arrows."

- Response: We agree and have adjusted Figure 5 accordingly.

Comment: "BD was found to be confusing throughout the manuscript due to its similarity with "binding domain" a suggestion would be to write out bromodomain or use Brd."

- Response: We thank the reviewer for pointing this out and have adjusted the abbreviation to BRD in the revised text.

Comment: "On line 384, the figure should be 8c and not 6c"

- Response: We thank the reviewer for noticing this and have corrected this in the revised text.

REVIEWERS' COMMENTS:

Reviewer #1 (Remarks to the Author):

Morrison et al presents data on the DNA binding activity of BRG1 and BRM bromodomains through a basic sequence patch formed by the α A helix and the ZA and AB loops. The author's suggest a bidentate binding mode of the BD recognizing Kac histone tails as well as DNA. The study reports interesting structural and biophysical data on this BD containing protein.

In the revision the authors created a mutant that should prevent DNA interaction in cells using gene editing techniques. However, the mutations that were introduced showed no functional consequences suggesting that Kac binding as well as DNA interaction are not relevant from BRG1/BRM recruitment to chromatin. The physiological role of the described DNA interaction remains therefore to be shown – The revised version of the paper does unfortunately not discuss these new data still highlighting the functional importance of DNA binding (e.g. in the discussion line 393 "...Instead, BRG1/BRM BRDs target nucleosomes through contacts with DNA" – this is certainly not an appropriate discussion of the available data. The interaction with DNA or potentially other nucleic acids (e.g. non-coding RNAs) could still be functional important but based on the data the authors describe in the rebuttal letter the DNA interaction observed in vitro does not seem to be important for BRG1/BRM recruitment to chromatin. The discussion around this topic should be therefore phrased more carefully throughout the paper.

My second concern was the discrepancy of affinities for Kac peptides reported in this study and in the literature – this deviations might be construct dependent but I don't think that they are the result of poor sample quality used by other labs. The deviations should be mentioned but since interaction with Kac peptides are not that relevant for this study, I don't see a major problem with these deviating values.

Reviewer #2 (Remarks to the Author):

The authors have done a nice job of addressing the comments for both sets of reviews. I just have a few more minor comments for how the arguments are presented.

1. On page 5 and page 7 the authors create a triple mutant to alter the basic patch on the protein which was shown to abrogate binding to DNA. This supports their argument, and is not unreasonable based on a simple electrostatic effect by EMSA. It would be helpful for the readers to note that the bromodomain is still well-folded and still able to bind histones as a needed control. Looking at the HSQC in the SI figure 8, it might simply be reasonable to add a sentence noting if the fold is intact based on the chemical shift fingerprint in the HSQC

2. On page 8, the statement "does not alter affinity" is not really correct as the difference between 0.76 and 1.12 look statistically different I would suggest putting the K_d 's in the figure 5C

3. On page 9, the authors cite the BrdT paper where the second domain binds DNA. However in their SI figure they calculate the pI of the second domain of BrdT to be slightly acidic (6.2) whereas the first domain is basic. Without further clarification this statement seems at odds with what they want to say.

We thank the reviewers for their thorough and thoughtful re-reading of our manuscript and their excellent additional suggestions. We believe we have addressed all additional comments/suggestions. Please see our response to specific comments below.

Reviewer 1

Comment: "In the revision the authors created a mutant....The revised version of the paper does unfortunately not discuss these new data...the DNA interaction observed in vitro does not seem to be important for BRG1/BRM recruitment to chromatin. The discussion around this topic should be therefore phrased more carefully throughout the paper."

- Response: We appreciate the concerns expressed by the reviewer, and agree that this needs to be more thoroughly and carefully discussed within the body of the text. In response, we have included the new data in Figure 6b and have made changes to the text to be more clear. In particular, we have clearly denoted discussion of data showing that DNA binding is important for nucleosome association in vitro, and that showing that it is not important for BRG1 chromatin recruitment.

Comment: My second concern was the discrepancy of affinities for Kac peptides reported in this study and in the literature – this deviations might be construct dependent but I don't think that they are the result of poor sample quality used by other labs. The deviations should be mentioned but since interaction with Kac peptides are not that relevant for this study, I don't see a major problem with these deviating values.

- Response: We have now more clearly denoted the discrepancy between affinities in the literature and those reported here on page 4.

Reviewer 2

Comment: On page 5 and page 7 the authors create a triple mutant to alter the basic batch on the protein which was shown to abrogate binding to DNA. This supports their argument, and is not unreasonable based on a simple electrostatic effect by EMSA. It would be helpful for the readers to note that the bromodomain is still well-folded and still able to bind histones as a needed control.

- Response: We agree that this is an important control. We have now included an HSQC titration of the H3K14ac peptide into the ¹⁵N-labeled bromodomain mutant in Supplementary Figure 8. This shows that the mutant is well folded and binds to histone peptide. This data is discussed on page 5.

Comment: On page 8, the statement "does not alter affinity" is not really correct as the difference between 0.76 and 1.12 look statistically different I would suggest putting the K_d's in the figure 5C

- Response: We agree and have altered the statement on page 8 to better reflect the data, and have included the K_d's in figure 5C.

Comment: On page 9, the authors cite the BrdT paper where the second domain binds DNA. However in their SI figure they calculate the pI of the second domain of BrdT to be slightly acidic (6.2) whereas the first domain is basic. Without further clarification this statement seems at odds with what they want to say.

- Response: We apologize for the mistake. It is, in fact, the first BRD of BrdT that was reported to bind to DNA and which we report to have a PI of 9.1. This mistake has been corrected in the text.